# PWWP2A binds distinct chromatin moieties and interacts with an MTA1-specific core NuRD complex

Stephanie Link[1,2], Ramona M.M. Spitzer[1,2], Maryam Sana[3], Mario Torrado[3], Moritz C. Völker-Albert[1], Eva C. Keilhauer[4,9], Thomas Burgold[5,10], Sebastian Pünzeler[1,11], Jason K.K. Low[3], Ida Lindström[3], Andrea Nist[6], Catherine Regnard[1], Thorsten Stiewe[6,7], Brian Hendrich[5], Axel Imhof[1,8], Matthias Mann[4,8], Joel P. Mackay[3], Marek Bartkuhn[2] & Sandra B. Hake[2,8]

Chromatin structure and function is regulated by reader proteins recognizing histone modifications and/or histone variants. We recently identified that PWWP2A tightly binds to H2A. Z-containing nucleosomes and is involved in mitotic progression and cranial–facial development. Here, using in vitro assays, we show that distinct domains of PWWP2A mediate binding to free linker DNA as well as H3K36me3 nucleosomes. In vivo, PWWP2A strongly recognizes H2A.Z-containing regulatory regions and weakly binds H3K36me3-containing gene bodies. Further, PWWP2A binds to an MTA1-specific subcomplex of the NuRD complex (M1HR), which consists solely of MTA1, HDAC1, and RBBP4/7, and excludes CHD, GATAD2 and MBD proteins. Depletion of PWWP2A leads to an increase of acetylation levels on H3K27 as well as H2A.Z, presumably by impaired chromatin recruitment of M1HR. Thus, this study identifies PWWP2A as a complex chromatin-binding protein that serves to direct the deacetylase complex M1HR to H2A.Z-containing chromatin, thereby promoting changes in histone acetylation levels.

[1] Department of Molecular Biology, BioMedical Center (BMC), Ludwig-Maximilians-University Munich, 82152 Planegg-Martinsried, Germany. [2] Institute for Genetics, Justus-Liebig University Giessen, 35392 Giessen, Germany. [3] School of Life and Environmental Sciences, University of Sydney, New South Wales 2006, Australia. [4] Department of Proteomics and Signal Transduction, Max Planck Institute of Biochemistry, 82152 Martinsried, Germany. [5] Wellcome Trust – MRC Stem Cell Institute and Department of Biochemistry, University of Cambridge, Cambridge CB2 1QR, UK. [6] Genomics Core Facility, Philipps-University Marburg, 35043 Marburg, Germany. [7] Institute for Molecular Oncology, Philipps-University Marburg, 35043 Marburg, Germany. [8] Center for Integrated Protein Science Munich (CIPSM), 81377 Munich, Germany. [9] Present address: Coriolis Pharma, Fraunhoferstr. 18B, 82152 Planegg, Germany. [10] Present address: Wellcome Sanger Institute, Wellcome Genome Campus, Hinxton Cambridge CB10 1SA, UK. [11] Present address: Coparion GmbH & Co. KG, Charles-de-Gaulle-Platz 1d, 50679 Cologne, Germany. These authors contributed equally: Stephanie Link, Ramona M.M. Spitzer. Correspondence and requests for materials should be addressed to M.B. (email: marek.bartkuhn@gen.bio.uni-giessen) or to S.B.H. (email: sandra.hake@gen.bio.uni-giessen.de)

In 2000, the so-called "histone code hypothesis" was proposed[1], bringing forward the concept that histone posttranslational modifications (PTMs) can be recognized by chromatin-modifying proteins through the action of "reader domains" that recognize specific histone PTMs. Such recognition events lead to the recruitment of proteins or protein complexes with enzymatic activity, which then act to alter the surrounding chromatin structure and consequently either promote or repress transcription of the associated gene(s). Furthermore, histone variants, which are deposited into chromatin through enzyme-catalyzed exchange with their canonical counterparts[2], have been found to codetermine specificity of coregulator binding, in this way contributing to the combinatorics of chromatin recognition.

Genome-wide, distinct chromatin states are defined by their specific histone mark pattern, and these patterns have been shown to modulate transcriptional output. For example, the presence of acetylation on either H3 lysine 27 (H3K27) or the histone variant H2A.Z correlates with regions identified as regulatory enhancers[3]. Unmodified H2A.Z is enriched at the transcriptional start site (TSS), where it is thought to promote transcription of the associated gene[4]. H3K36me3 is present at active genes towards the 3′-end and is likewise involved in transcriptional regulation[5]. The dynamic interplay of chromatin states is established and controlled in part by protein complexes such as the nucleosome remodeling and deacetylase complex (NuRD), which typically combines both chromatin remodeling and histone deacetylase activities[6]. The canonical NuRD complex comprises at least six different proteins: HDAC1 and/or HDAC2 which mediate histone deacetylase (HDAC) activity; CHD3 or CHD4, which remodel nucleosomes in an ATP-dependent manner; MTA1–MTA3; MBD2/MBD3; RBBP4/RBBP7; and GATAD2A/GATAD2B. Interestingly, different combinations of the NuRD subunits are able to form mutually exclusive NuRD subcomplexes[7–9]. First described as a transcriptional silencer, emerging evidence attributes a broader functional spectrum to the NuRD complex including transcriptional activation or balancing transcriptional output[10,11].

Recently, we identified the vertebrate-specific PWWP2A as a chromatin-binding transcriptional regulator that mediates proper mitosis-progression in human cell lines and correct cranial–facial development in the frog via a yet unknown mechanism[12]. PWWP2A contains two N-terminal proline-rich regions (P1 and P2) connected through an unique internal region (I) to a serine-rich stretch (S) and a C-terminal PWWP domain (PWWP). Strong chromatin interaction of PWWP2A is achieved through binding to nucleosomes in general via the N-terminal part of the internal stretch (IN), whereas the C-terminal internal region (IC) mediates specificity for H2A.Z-containing nucleosomes. In addition, the conserved PWWP domain is able to bind DNA with relatively low affinity[12].

Using quantitative in vitro binding assays, here we show that PWWP2A is able to directly interact with at least five different chromatin moieties with its distinct domains. Besides strong binding to H2A.Z and the TSS of highly transcribed genes via its IC region[12], IN directly interacts with linker DNA, a combination of S and PWWP domains mediates binding to H3K36me3 as well as nucleic acids in vitro and promotes weak association with H3K36me3-containing gene bodies in vivo. Additionally, PWWP2A is also enriched at H2A.Z-containing regulatory regions marked by high levels of both H3K27ac and H2A.Zac. Quantitative mass spectrometry (MS) and biochemical characterization identifies a robust interaction between PWWP2A and a subcomplex of NuRD comprising MTA, HDAC and RBBP proteins (MHR). We demonstrate that the P1_P2 region of PWWP2A specifically requires binding to the MTA1 isoform for recruitment of the entire MHR complex (M1HR). In agreement with our model of PWWP2A serving as an adapter between H2A.Z chromatin and histone modifying complexes (such as HDACs), small interfering RNA (siRNA)-mediated depletion of PWWP2A results in hyperacetylation of H3K27 and H2A.Z at some PWWP2A-marked regulatory regions. Overall, our data demonstrate that PWWP2A recruits the M1HR complex to H2A.Z-containing chromatin, providing a mechanism by which PWWP2A cooperates with H2A.Z to regulate chromatin structure.

## Results

### PWWP2A's internal region binds H2A.Z nucleosomes and DNA.
Recently, we showed that the internal region (I) of PWWP2A, which does not exhibit any sequence homology to other proteins/protein domains, facilitates binding of PWWP2A to mononucleosomes in general via its N-terminal part (IN), and that its C-terminal section (IC) directs specificity for H2A.Z-containing nucleosomes[12]. To determine whether IN or IC are able to directly bind to H2A.Z-containing mononucleosomes or whether this interaction is indirectly mediated by additional PWWP2A-associated factors or posttranslational histone modifications, we carried out in vitro binding assays. We assembled mononucleosomes with 187-bp DNA (Widom 601-sequence[13]) and recombinant histone octamers containing either H2A.Z or H2A (Supplementary Fig. 1a). The two types of nucleosomes incorporated distinct fluorescent tags on the DNA, allowing us to prepare 1:1 mixtures of H2A and H2A.Z nucleosomes and perform competitive electrophoretic mobility shift assays (cEMSAs) using purified recombinant GST-I, GST–IN, or GST-IC proteins (Fig. 1a and Supplementary Fig. 1b). GST-I bound to both H2A.Z and H2A nucleosomes, but with a higher preference for H2A.Z nucleosomes (Fig. 1b). In line with our previous results using cell-derived mononucleosomes[12], IN bound both recombinant H2A and H2A.Z nucleosomes equally well, while the IC region showed a clear preference for H2A.Z-containing nucleosomes (Fig. 1b).

Next, we asked whether the nonspecific IN-nucleosome interaction is mediated either through the histones and/or the DNA itself. As most proteins recognize the flexible histone tails rather than the buried histone core regions[14], we tested a series of recombinant nucleosomes containing flexible tail deleted histones (tail-less, TL) (Supplementary Fig. 1c) in cEMSAs. TL nucleosomes did not result in less IN binding, suggesting that the histone tails themselves are not needed for IN-nucleosome binding. Interestingly, at the highest protein concentration (80 nM), two shifted bands are present, indicating that each mononucleosome can presumably be bound by two GST–IN proteins as two DNA linkers are available. Further, we noticed that in the presence of linker DNA TL H3 nucleosomes were bound much better by IN than WT nucleosomes (Fig. 1c). This observation is in line with published reports showing H3 tail interacting with linker DNA[15,16], thereby possibly preventing efficient IN recruitment. On the other hand, linker-less TL H4 containing nucleosomes also led to a slight increase in IN binding (Supplementary Fig. 1d), possibly due to the H4 tail not blocking the acidic patch of H2A, as published previously[17]. IN has no sequence homology with other DNA-binding motifs but is very conserved between amphibians and mammals (Supplementary Fig. 1e). To test whether IN could be a possible DNA-binding domain, we performed EMSAs with free DNA (187 bp) and observed that IN, but not IC strongly interacts with nucleic acids (Fig. 1d).

Next, we tested whether DNA binding is the sole mediator of nucleosome binding and whether IN can recognize histone-wrapped nucleosomal DNA as well as free DNA. We performed cEMSAs with recombinant H2A.Z- (Fig. 1e) or

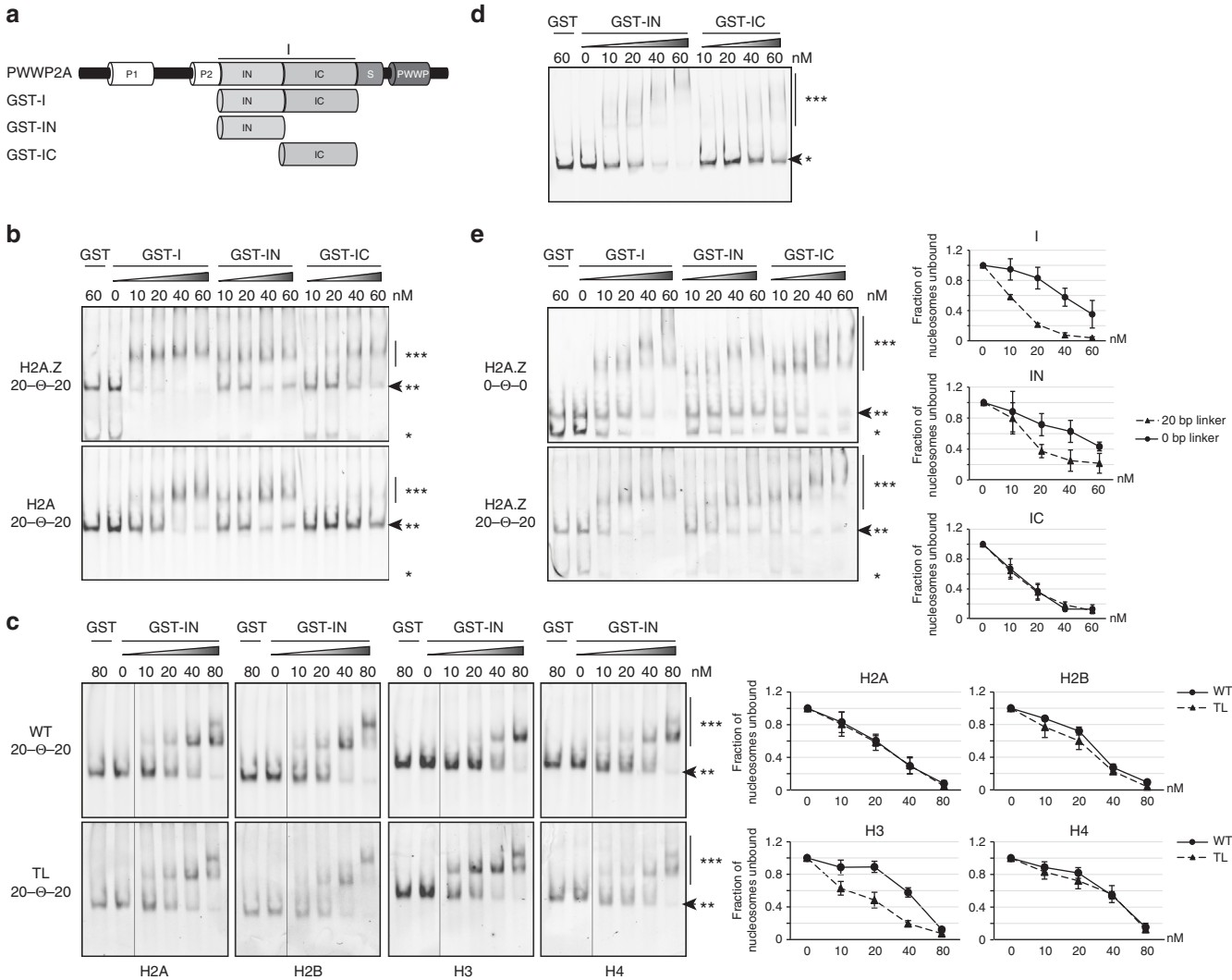

**Fig. 1** IC distinguishes between H2A and H2A.Z, whereas IN recognizes nucleosomal linker DNA. **a** Schematic representation of recombinant GST–PWWP2A deletions (GST-I, GST-IN, and GST-IC) used in cEMSAs. **b** Representative cEMSA in which a 1:1 mixture of H2A- and H2A.Z-containing nucleosomes (each with a distinct fluorescent tag) was incubated with increasing concentrations of GST-tagged I-domain constructs. The top gel shows detection of the H2A.Z nucleosomes and the bottom gel detection of the H2A nucleosomes. In both cases, the DNA contained a 20-bp linker DNA (Widom 601-sequence) on each side of the nucleosome (20–Θ–20). GST alone served as negative control. *Free DNA, **nucleosome, ***nucleosome GST–protein complex. Arrow indicates loss of signal when nucleosome GST–protein complexes are formed. **c** Left: representative cEMSAs similar to (**b**) using recombinant wildtype mononucleosomes (WT, top) or mononucleosomes lacking single histone tails (TL, bottom) containing 20-bp linker DNA (20–Θ–20). Nucleosomes were incubated with the indicated concentrations of GST–IN and the gel visualized by fluorescence detection of the indicated nucleosome. Right: quantification of signal intensities of nucleosomes (**) using Image Studio Lite Ver 5.2 (LI-COR). Error bars indicate SEM of three independent replicates. **d** Representative EMSA using Cy-5 labeled 187-bp dsDNA and the indicated concentrations of GST–IN and GST–IC. *free DNA, ***DNA–GST–protein complex. Arrow indicates unbound DNA. **e** Left: representative cEMSAs similar to (**b**) using recombinant H2A.Z-containing mononucleosomes without (0–Θ–0, top) and with (20–Θ–20, bottom) linker DNA; these nucleosomes were incubated with the indicated concentrations of GST-I, GST-IN, and GST-IC. *Free DNA, **nucleosome, ***nucleosome GST–protein complex. Arrow indicates loss of signal when nucleosome GST–protein complexes are formed. Right: Quantification of signal intensities of nucleosomes (**) using Image Studio Lite Ver 5.2 (LI-COR). Error bars indicate SEM of three independent replicates

H2A-nucleosomes (Supplementary Fig. 1f) that either displayed or lacked linker DNA sequence on either side of the Widom sequence. In the case of H2A.Z nucleosomes, binding was observed for all three PWWP2A constructs (I, IN, and IC) irrespective of the presence of linker DNA; however, both I and IN constructs showed stronger binding to nucleosomes containing the linker sequence. In the case of H2A nucleosomes, binding was much weaker overall but again a preference was observed for linker-containing nucleosomes. As expected, IC's nucleosome binding ability depended almost exclusively on the presence of H2A.Z and not on the DNA length.

Our data indicate that the IC region of PWWP2A preferably interacts with H2A.Z-containing nucleosomes, and that the IN region binds with a preference for free/linker DNA.

**PWWP2A's PWWP domain recognizes H3K36me3**. PWWP2A also contains an evolutionary conserved PWWP domain, which is characterized by a proline–tryptophan–tryptophan–proline motif that has been shown to mediate DNA and/or histone H3 lysine 36 trimethylation (H3K36me3) binding in other proteins[18,19]. We, therefore, tested whether the PWWP2A PWWP domain is also

able to recognize one or even both chromatin features directly. Previously, we detected low-affinity binding of PWWP2A's PWWP domain to free DNA[12], but did not investigate whether this domain is able to recognize histone-bound nucleosomal DNA as well. cEMSAs with purified recombinant GST–PWWP (Supplementary Fig. 2a) and recombinant H2A.Z or H2A nucleosomes with or without linker DNA (Supplementary Fig. 2b, c) clearly showed the strong dependence of PWWP on free linker DNA for efficient nucleosome binding (Fig. 2a). Interestingly, the two observed upshifted bands suggest that two PWWP domains might bind to one nucleosome simultaneously. As reported for other PWWP-domain containing proteins (reviewed in ref. [19]), this interaction appears to be only charge

mediated, as the PWWP domain binds equally well to different types of single and double stranded nucleic acids (Supplementary Fig. 2d). Comparison of DNA-binding abilities of GST–IN, and GST–PWWP via EMSA showed that both domains are able to bind DNA, albeit with different affinities; IN shows a higher binding affinity than PWWP at 100 and 200 nM, respectively (Supplementary Fig. 2e).

PWWP domains are characterized by several highly conserved aromatic cage residues (Supplementary Fig. 2f) that, in other proteins, have been shown to mediate recognition of methylated histones, most commonly trimethylated H3 lysine 36 (H3K36me3)[20]. The PWWP2A PWWP domain contains such amino acids (F666, W669, and W695) that when modeled in

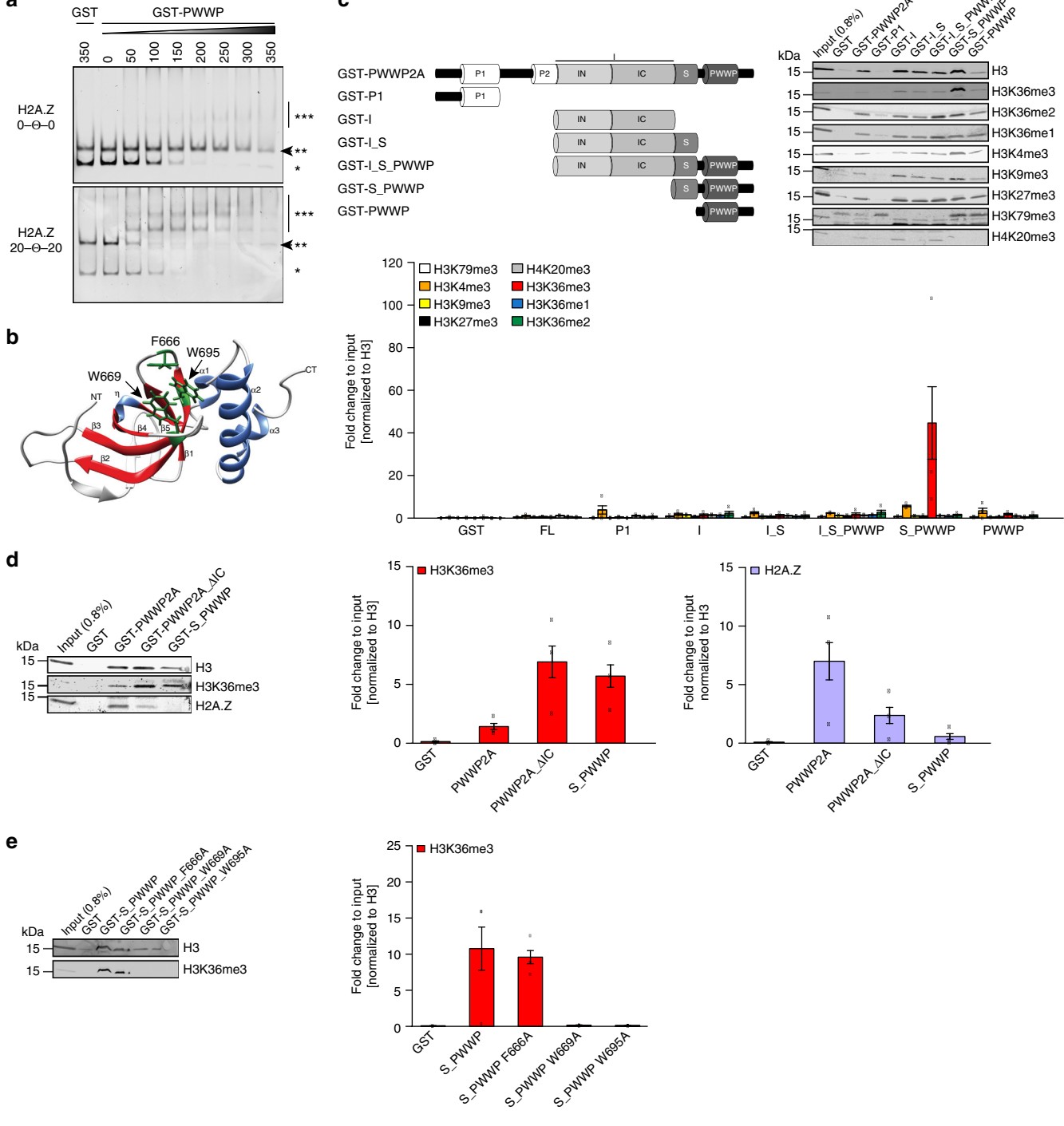

silico are proposed to form an aromatic cage (Fig. 2b). To investigate whether PWWP2A can recognize PTMs on histone tails, we incubated recombinant PWWP2A or its truncations with micrococcal nuclease (MNase)-digested mononucleosomes derived from HeLa Kyoto (HK) cells and investigated PTM enrichment in immunoblots (Fig. 2c). Interestingly, only the combination of the serine-rich stretch with PWWP domain (S_PWWP), but not other PWWP2A deletion constructs efficiently pulled down nucleosomes enriched in H3K36me3 (Fig. 2c). Deletion of the H2A.Z interaction domain IC (PWWP2A_ ΔIC) led to an increased binding to H3K36me3 from cell-derived mononucleosomes while binding to H2A.Z was dramatically reduced (Fig. 2d). These data indicate that H2A.Z binding to the IC region is preferred over H3K36me3 binding to S_PWWP.

The interaction of S_PWWP with H3K36me3 nucleosomes depended on a functional aromatic cage, as point mutations of two of the three aromatic cage residues completely abolished the interaction with H3K36me3 (Fig. 2e) (without affecting folding, as determined by nuclear magnetic resonance (NMR) spectrometry; Supplementary Fig. 2g). An MS version of the pulldown assay and subsequent gel-excision of histone bands after IP (Supplementary Fig. 3a) independently verified the enrichment of H3K36me3 nucleosomes in pulldowns with S_PWWP (Supplementary Fig. 3b, Supplementary Data 1). In conclusion, the PWWP2A PWWP domain mediates interactions with chromatin via nucleic acid binding as well as aromatic cage-dependent H3K36me3 recognition.

**PWWP2A binds H3K36me3 gene bodies and regulatory regions**. Our in vitro data show that PWWP2A strongly interacts with H2A.Z as well as linker DNA-containing nucleosomes using its internal domain and is also able to recognize H3K36me3 with its PWWP domain (Supplementary Fig. 4a). We used our previously generated GFP–PWWP2A nChIP-seq data[12] to ask whether these in vitro observations are also supported in vivo. We first searched for gene body regions with high H3K36me3 levels (ENCODE) by cluster analysis, identifying cluster 2 as prototypic for H3K36me3-enriched gene body peaks (Fig. 3a). Meta-gene profile comparison of cluster 2 genes with all other genes verified the strong enrichment of both PWWP2A and H2A.Z at these promoters, and also revealed a slight increase in PWWP2A signal at H3K36me3-positive gene bodies of cluster 2 genes. In contrast, both H2A.Z isoforms showed a depletion in the gene body regions (Fig. 3b). Comparison of H3K36me3, GFP–PWWP2A, and GFP-H2A.Z signals in 3′-gene body ends with the

corresponding non-transcribed downstream regions revealed a small but significant enrichment of PWWP2A across H3K36me3-high gene bodies (Fig. 3c and Supplementary Fig. 4b). Reciprocally, H2A.Z.1 and H2A.Z.2 were depleted from H3K36me3-positive gene bodies (Fig. 3c and Supplementary Fig. 4b), showing that PWWP2A recruitment to those sites is independent of H2A.Z and likely occurs only in a transient manner.

Recently, we have described PWWP2A as a strong interactor of H2A.Z, irrespective of whether the latter is located at the TSS or at other sites in the genome[12]. To determine whether PWWP2A is also enriched at other chromatin features/states, we reanalyzed our recently published nChIP-seq data[12]. We compared our data to chromatin states defined by training a 10-state model on ENCODE data for H3K4me3, H3K4me1, H3K27ac, H3K36me3, and H3K27me3 using ChromHMM[21,60]. PWWP2A binding was strongly enriched at active promoters, which are characterized by high levels of H3K4me3 and H3K27ac (state 3), as well as active and inactive/poised enhancers, which are marked by high levels of H3K4me1 and high or low H3K27ac, respectively (states 4 and 5) (Fig. 4a–c). The enrichment pattern observed for H2A.Z was very similar to that of PWWP2A, indicating a common role for both factors at respective chromatin regions. Intriguingly, H2A.Z and PWWP2A appear to co-occupy enhancer sites as well (clusters 3 and 4), suggesting a mechanism by which PWWP2A is recruited to regulatory regions (Fig. 4c).

Overall, these data show that our in vitro observations regarding the interactions made by PWWP2A with H2A.Z and H3K36me3 are recapitulated in vivo and further reveal a connection of PWWP2A with regulatory regions.

**PWWP2A interacts preferentially with an MTA1-specific core NuRD complex (M1HR)**. Having determined that PWWP2A mediates a range of interactions with chromatin via its distinct domains, we next sought to define the mechanism by which PWWP2A affects transcriptional regulation[12]. Because PWWP2A does not contain any domains with known enzymatic activity, we speculate that it recruits chromatin modifiers to H2A.Z nucleosomes, and possibly to other H3K36me3-enriched sites. We, therefore, searched for PWWP2A binding partners using the same assay that we used previously to determine the H2A.Z interactome[12,22]. Briefly, chromatin isolated from HK cells stably expressing GFP or GFP–PWWP2A were digested with MNase to mononucleosomes (Supplementary Fig. 5a) and immunoprecipitated with GFP-TRAP beads. GFP- or GFP–PWWP2A bound nucleosomal proteins were subjected to on-bead tryptic digestion

**Fig. 2** PWWP domain binds nucleic acids and S_PWWP interacts with H3K36me3. **a** Representative cEMSA using recombinant H2A.Z-containing mononucleosomes assembled either without (0–Θ–0, top) or with linker DNA (20–Θ–20, bottom) incubated with indicated increasing concentrations of GST–PWWP. GST alone served as negative control. *Free DNA, **nucleosome, ***nucleosome GST–PWWP complex. Arrow indicates loss of signal when nucleosome GST–protein complexes are formed. **b** In silico structure of PWWP domain modeled with the web browser-based tool iTASSER and visualized with Chimera (1.8.0). β-barrels (β1–β5) are colored in red, α-helixes (α1–α3), and η-helix in blue and the three residues forming the aromatic cage (F666, W669, and W695) are highlighted in green and depicted in stick mode. NT = N-terminus, CT = C-terminus. **c** Top left: schematic representation of recombinant GST–PWWP2A and deletions (GST–P1, GST-I, GST-I_S, GST-I_S_PWWP, GST-S_PWWP, and GST–PWWP) used in cell-derived mononucleosome-IPs. Top right: Immunoblotting of different histone PTMs upon GST–PWWP2A deletion construct (GST–PWWP2A, GST-P1, GST-I, GST-I_S, GST-I_S_PWWP, GST-S_PWWP, and GST–PWWP) IPs with HK cell-derived mononucleosomes. Notice enrichment of H3K36me3 in comparison to other modifications in S_PWWP pulldown. GST alone served as negative control. Bottom: Data quantification was done for three biological replicates for each PTM (n = 3). Data shown are means and error bars depict SEM. **d** Left: immunoblotting of H3K36me3 and H2A.Z upon GST–PWWP2A, GST–PWWP2A_ΔIC and GST-S_PWWP IPs with HK cell-derived mononucleosomes. Right: Data quantification of H3K36me3 enrichment (middle) and H2A.Z binding (right) was done for three biological replicates (n = 3). Data shown are means and error bars depict SEM. **e** Left: immunoblotting of H3K36me3 upon GST-S_PWWP aromatic cage point mutants (GST-S_PWWP_F666A, GST-S_PWWP_W669A, GST-S_PWWP_W695A) IPs with HK cell-derived mononucleosomes. Notice reduction of H3K36me3 in GST-S_PWWP_W669A and GST-S_PWWP_W695A pulldowns. Right: Data quantification was done for three biological replicates (n = 3). Data shown are means and error bars depict SEM

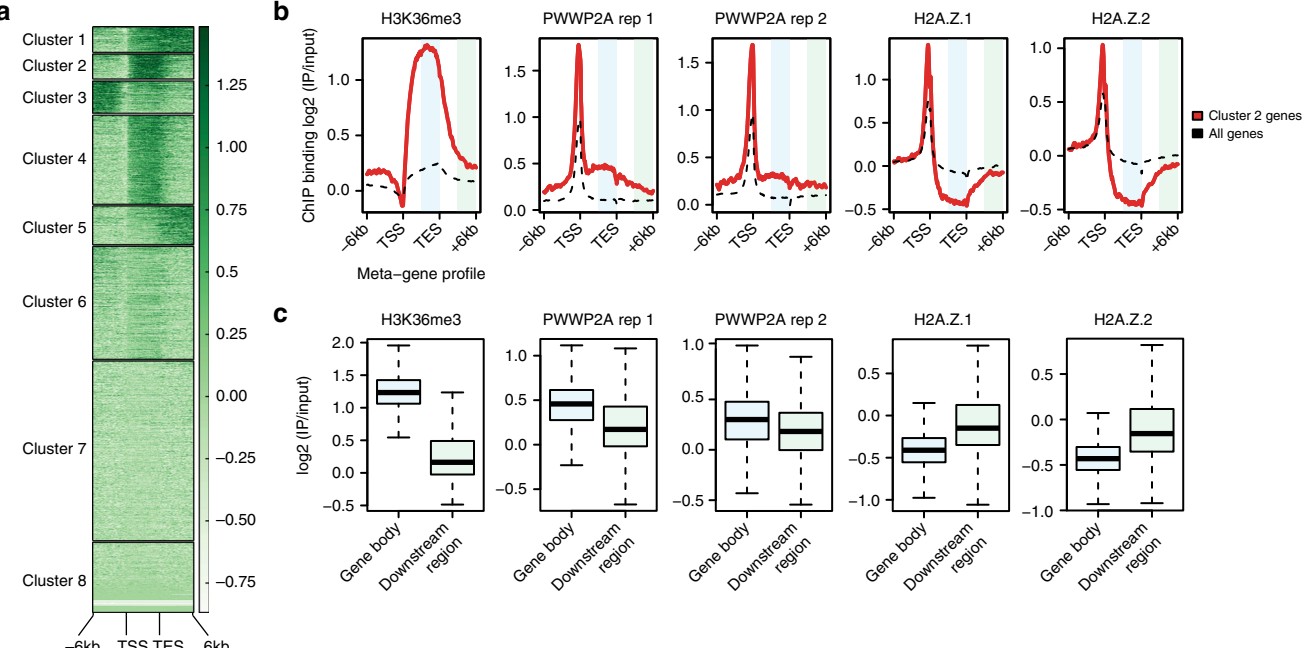

**Fig. 3** PWWP2A associates weakly with H3K36me3-enriched, H2A.Z-depleted gene body regions. **a** ChIP-seq density heatmap of H3K36me3 (ENCODE) clustering analysis of meta-gene binding profiles encompassing gene bodies and 6 kb upstream and downstream regions of transcriptional start sites (TSS) and transcriptional end sites (TES). Color intensity represents normalized and scaled tag counts. Cluster 2, which represents H3K36me3-enriched genes, was used for further analysis. **b** Meta-gene profile correlations of ChIP-seq data for H3K36me3, two GFP–PWWP2A replicates (rep), GFP-H2A.Z.1, and GFP-H2A.Z.2 mean coverage signals of cluster 2 genes (red line) and all genes (black dotted line) encompassing gene bodies and 6 kb upstream and downstream regions of TSS and TES. Regions highlighted in light blue (gene body) and light green (downstream region) were further analyzed in (**c**). GFP–PWWP2A and GFP-H2A.Z nChIP-seq data are from[12]. (**c**) Boxplots of H3K36me3, two independent GFP–PWWP2A, GFP-H2A.Z.1, and GFP-H2A. Z.2 signal intensities comparing gene body with non-coding regions of same size within cluster 2

and then quantitated by label-free MS/MS. Identification of H2A. Z as one of many PWWP2A targets provided confidence in the approach (Fig. 5a and Supplementary Fig. 5b). We detected several proteins and subunits of protein complexes previously identified to also interact with H2A.Z (e.g., BAHD1, BRWD3, HMG20A, PHF14, PHF20L1, RAI1, and ZNF512B) and some that appear to be PWWP2A-specific (e.g., ATRX, DAXX, MDC1, and PWWP2B). Interestingly, among the proteins that were previously shown to precipitate with H2A.Z, strongest binding and collective clustering in volcano plot (Fig. 5a) was observed for MTA1, HDAC2, RBBP4, and RBBP7, which are core components of the nucleosome remodeling and deacetylase (NuRD) complex that is involved in gene regulation[23], and implicated in cancer and various other diseases[6]. PWWP2A binding to HDAC1/2 has been previously reported[24], further corroborating our data. Usually, NuRD also contains MBD2/3, GATAD2A/B and the ATP-dependent remodeling enzymes CHD3/4/5[6]. Surprisingly, none of these subunits were isolated in our PWWP2A (Fig. 5a, b, and Supplementary Fig. 5b) or H2A.Z pulldowns[12], suggesting that the MTA, HDAC and RBBP subunits constitute an independently stable subcomplex that we named MHR after its constituents. A corresponding PMR complex has been identified in *Drosophila*[25], which contains the three homologous proteins (p55, MTA-like, and Rpd3). First, we wondered whether PWWP2A is able to equally interact with all MTA isoforms, which differ mainly in their C-terminal regions (Fig. 5c). We performed tagged MTA isoform-specific IPs followed by immunoblotting for PWWP2A and, as positive control, MBD3. We found that PWWP2A showed strongly enriched binding to MTA1 over MTA2 or MTA3 (Fig. 5c). Similarly, in a reciprocal experiment, only MTA1 but not MBD3 was able to efficiently pulldown PWWP2A (Supplementary Fig. 5c), confirming MTA1–PWWP2A

association without MBD3 presence. To determine which subunit (s) of the MHR complex is responsible for mediating the interaction with PWWP2A, a FLAG-PWWP2A IP was performed with cell lysates from HEK293 cells coexpressing FLAG-PWWP2A, HDAC1 (untagged), HA-RBBP4 and either HA-MTA1 or HA-MTA2 (Fig. 5d). Remarkably, only in the presence of MTA1, but not MTA2, PWWP2A pulled down all components of the MHR complex, suggesting that MTA1 is the direct binding partner of PWWP2A and mediates the recruitment of HDAC and RBBP proteins to form an MTA1-specific M1HR module. We hypothesize that PWWP2A might compete with the MBD–GATA–CHD (MGC) subunits for binding to the M1HR module, explaining why MBD, GATA and CHD proteins are not enriched in PWWP2A purifications.

We next assessed which region(s) of PWWP2A is responsible for mediating the MTA1–PWWP2A interaction. We incubated recombinant GST-tagged PWWP2A deletions with lysate from HEK293 cells expressing full-length MTA1-HA. The strongest interaction was observed with a fragment containing the P1, P2, and I regions of PWWP2A, and a weaker interaction was observed with the P1 domain alone (Fig. 5e). The importance of the P1_P2_I region for M1HR recruitment was also confirmed for endogenous MTA1 and HDAC2 when using recombinant GST–PWWP2A deletions incubated with HK cell-derived mononucleosomes (Supplementary Fig. 5d). As neither the I or P1 domain alone nor the P2_I region were able to bind MTA1, we speculate that the P2 domain contributes the most to the interaction with MTA1, most likely with some contribution from the P1 domain.

In summary, we have identified a specific and direct interaction of PWWP2A with a core M1HR complex that is mediated via the P1_P2 region in PWWP2A.

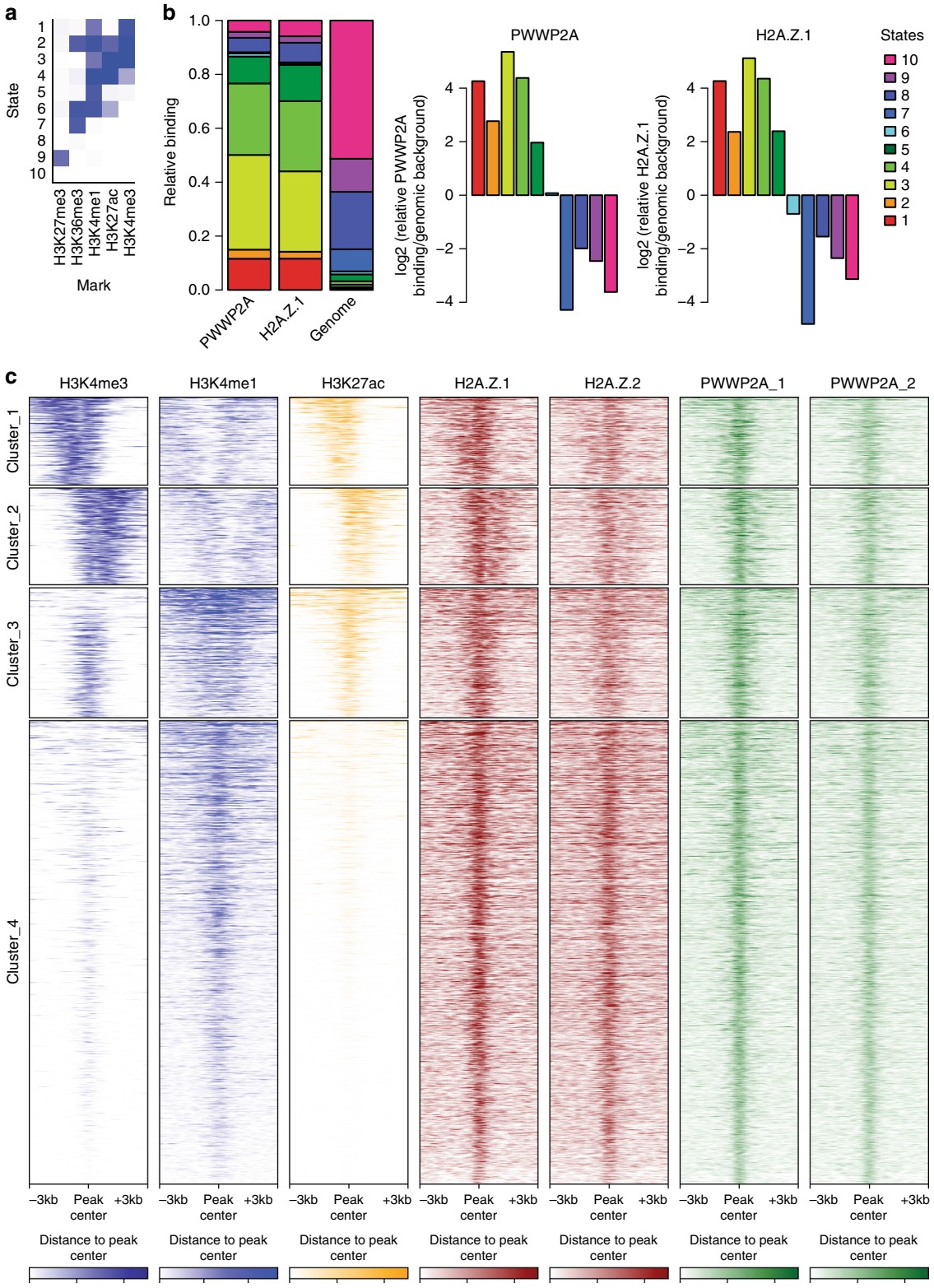

**PWWP2A loss affects histone acetylation at regulatory sites.** PWWP2A not only co-localizes with H2A.Z at the TSS and to a lesser extent with H3K36me3-enriched gene bodies, but also co-occupies regulatory regions marked by high H3K27ac levels (Fig. 4c). Previously, it was shown that H3K27ac levels correlate with activity of the NuRD complex[26]. The PWWP2A-associated M1HR complex that we identified contains the enzymatically active histone deacetylase HDAC2 that was also found to bind to H2A.Z-containing nucleosomes[12], as well as HDAC1. HDACs are generally believed to act as transcriptional corepressors by removing transcription-promoting histone acetylation marks[27,28]. However, recent experiments showed a strong association of all HDAC classes with active genes, in particular at gene promoters and enhancers[29,30]. These data sets nicely correlate with our

**Fig. 4** PWWP2A associates with regulatory regions genome-wide. **a** ChromHMM[21,60]-based characterization of chromatin states of PWWP2A-containing genomic regions. The heatmap depicts the emission parameters of the HMM and describes the combinatorial occurrence of the individual histone modifications in different chromatin states. **b** Chromatin-state enrichment of PWWP2A- or H2A.Z.1-enriched sites compared to complete human genome (left) and log2-fold enrichment or depletion of PWWP2A sites in specific states calculated to frequency in complete genome (right). Notice enrichment of PWWP2A in states 1–5 that resemble promoter and enhancer regions. **c** nChIP-seq density heatmap of PWWP2A enriched sites (two replicates, green) and visualization of the intensities of H3K4me3 (dark blue, ENCODE), H3K4me1 (light blue, ENCODE), H3K27ac (yellow, ENCODE), and H2A.Z1 and H2A.Z.2 variants (red) at these regions. Color intensity represents normalized and globally scaled tag counts. Notice PWWP2A is found at active promoter regions (H3K4me3+, H3K4me1−, H3K27ac+) encompassing clusters 1 and 2, as well as active (cluster 3: H3K4me3+, H3Kme1+, H3K27ac+) and inactive (cluster 4: H3K4me3−, H3K4me1+. H3K27ac−) enhancers, all containing H2A.Z variants

observation that HDAC2 and HDAC1 are bound to H2A.Z and PWWP2A, which both localize to the well-positioned −1 and +1 nucleosomes at the TSS of actively transcribed genes[12] and regulatory regions (Fig. 4c). Hence, we next wondered whether the H2A.Z–PWWP2A–M1HR connection is able to affect histone acetylation levels. As H3K27ac shares a substantial number of binding sites across the genome with acetylated H2A.Z (H2A.Zac, which can be acetylated at lysines 4, 7, and 11) at active TSS and enhancers (Supplementary Fig. 6), we analyzed H3K27ac as well as H2A.Zac levels after PWWP2A knockdown via nChIP-seq. Focusing on genome-wide distribution patterns of our H3K27ac or H2A.Zac nChIP-seq data indicated that the vast majority of sites were co-occupied by PWWP2A (Fig. 4c, Fig. 6a, Supplementary Fig. 6a, b). Strikingly, loss of PWWP2A led to a strong increase of H3K27ac and H2A.Zac levels at a relatively small, but reproducibly detectable fraction of 566 sites typically bound by PWWP2A and H2A.Z (Fig. 6a, b, Supplementary Fig. 7). Stringent data analysis revealed that this differential increase of H3K27 acetylation levels after PWWP2A depletion at these sites was likewise apparent at the majority of H2A.Zac sites and vice versa (Fig. 6b). The increase in acetylation predominantly occurred at distal intergenic regions (Supplementary Fig. 6c). To test whether the increase in acetylation on H3K27ac and H2A.Zac affects the expression of nearby genes, we analyzed expression levels via real-time quantitative polymerase chain reaction (RT-qPCR) and performed nChIP-qPCR to validate nChIP-seq results from corresponding regulatory regions (Fig. 7c). We found that the highly reproducible differential acetylation on H3K27ac as well as H2A.Zac led to the upregulation of expression of some genes (e.g., CCL5 and FST), while the expression levels of other genes remained unchanged (e.g., ZNF19 and B3GALNT2) (Fig. 6c, Supplementary Fig. 7).

Our data suggest that PWWP2A serves as an adapter between H2A.Z chromatin and the HDAC-containing M1HR complex, promoting in some cases changes in histone acetylation levels.

**Discussion**

We have identified PWWP2A as a multivalent, high-affinity chromatin binder that directly interacts with at least five different chromatin moieties (free DNA, histones, H2A.Z, H3K36me3, and nucleic acids) by utilizing at least four different domains (IN, IC, S_PWWP, and PWWP) (Supplementary Fig. 4a and Fig. 7). The functional consequences of PWWP2A's capability to selectively and/or synchronously bind distinct genomic sites and to recruit several chromatin-modifying proteins, such as the M1HR complex, are potentially complex. While direct H2A.Z nucleosome binding via the internal region appears to be responsible for PWWP2A's recruitment to enhancers and promoters of highly active genes, interaction of the S_PWWP domain with H3K36me3 is functionally more enigmatic. It has been noted for several other proteins, including ZMNYD11 that the PWWP domain alone is not sufficient for chromatin recruitment and a combination of the PWWP domain with an

adjacent domain is needed for proper binding of H3K36me3[31]. Although the recognition of this PTM, which is found at gene bodies of transcribed genes and is involved in suppressing cryptic transcript initiation[32,33], is observed for recombinant S_PWWP, constructs that include the H2A.Z-specific internal region—including full-length PWWP2A—are unable to bind H3K36me3 efficiently. We speculate that H2A.Z binding to the internal region of full-length PWWP2A is preferred over a PWWP2A–H3K36me3 association. This hypothesis is supported by our findings that PWWP2A lacking the H2A.Z-specificity module IC loses H2A.Z–nucleosome interaction, while at the same time it shows enhanced binding to H3K36me3-containing nucleosomes. How differential chromatin-state recognition by PWWP2A is regulated and possibly 'switchable' is not clear. We postulate that either IC binding to H2A.Z–nucleosomes dominates over S_PWWP–H3K36me3 interaction or that an element in the I domain might auto-inhibit the S_PWWP domain. The latter mechanism could be regulated by dynamic covalent modifications of PWWP2A that modulate its ability to bind these two mutually exclusive chromatin moieties. Intriguingly, the I domain harbors a number of sites that are potentially subject to lysine acetylation and also several potential phosphorylation sites. Further work will be required to test the hypothesis that modifications of PWWP2A in response to external signals might regulate its binding activity. Nevertheless, our finding that PWWP2A is able to bind distinct chromatin moieties is in line with the observation that full-length PWWP2A protein is mainly found at H2A.Z-containing promoter nucleosomes in nChIP-seq and that only a minor fraction is found at H3K36me3-positive gene bodies.

In line with its putative role as a transcriptional coregulator, we predicted that PWWP2A would bind to a range of chromatin-modifying proteins and complexes. Affinity purification of tagged PWWP2A from HK cells revealed a number of significantly enriched proteins, most of which were identified previously in an H2A.Z pulldown[12], several others appeared to be PWWP2A-specific, such as PWWP2B, MDC1, DAXX and ATRX. Three of the six core components of the NuRD complex—the MTA, HDAC, and RBBP subunits—were among the most enriched binding partners for PWWP2A in our affinity purification. Surprisingly, the MBD, GATAD2, and CHD subunits, which account for the chromatin remodeling and methyl-DNA binding functions of the NuRD complex, were not observed at significant levels.

Although several forms of the NuRD complex have been observed previously, these variants have largely been confined to paralogue switching—MBD2 versus MBD3, for example, or the inclusion of additional subunits such as CDKAP1[8,9,34]. Zhang et al.[25] reported that they were able to isolate a stable and catalytically active complex from Drosophila cells that is the equivalent of our M1HR complex, although they did not demonstrate that this complex had a function in vivo. We recently showed that CHD4 is a peripheral component of the NuRD complex and our data, together with the findings of Zhang

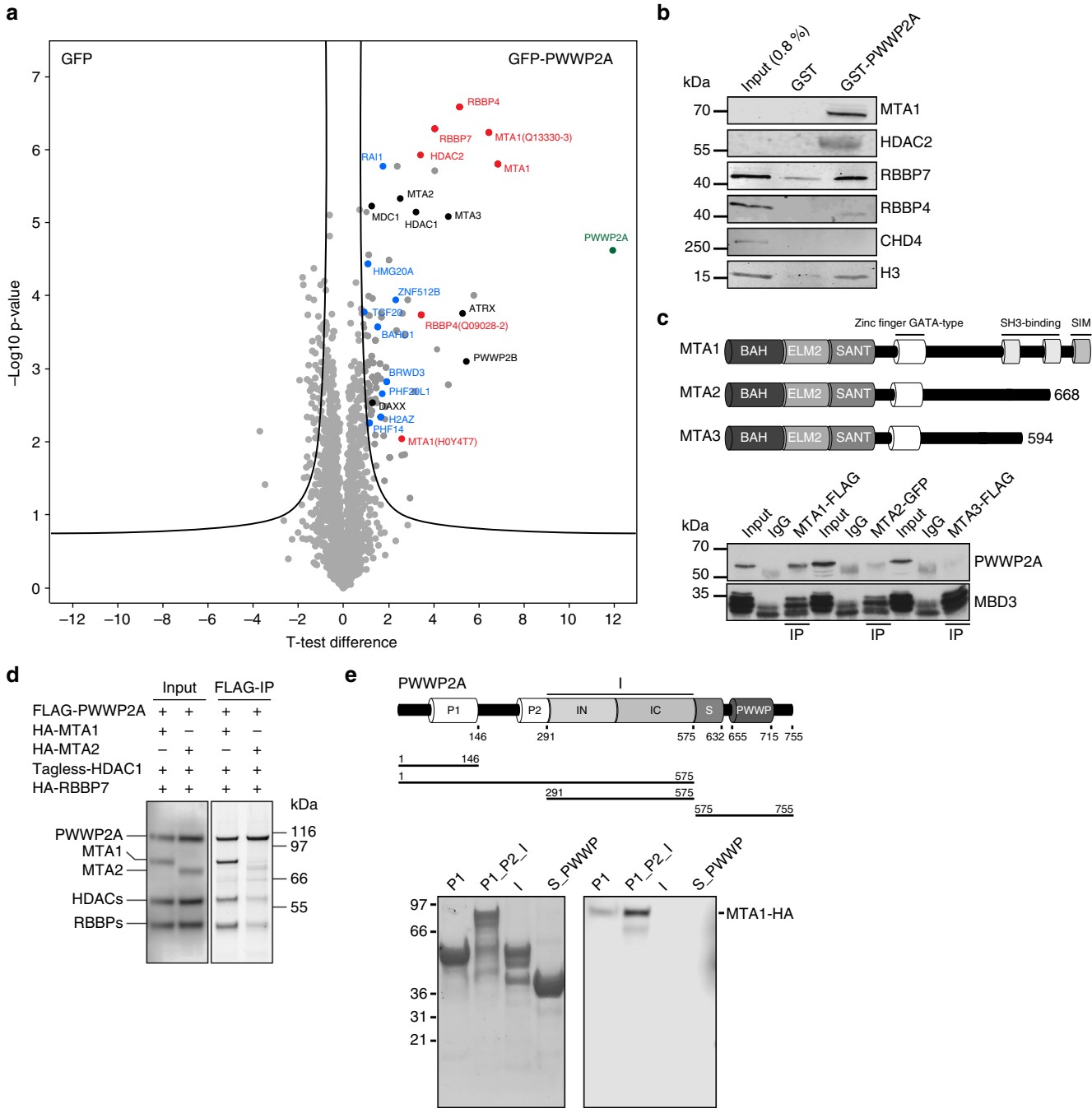

**Fig. 5** PWWP2A interacts with members of a core NuRD complex via MTA1. **a** Volcano plot of label-free interaction of GFP–PWWP2A-associated mononucleosomes. Significantly enriched proteins over GFP-associated mononucleosomes are shown in the upper right part. *t* test differences were obtained by two-sample *t* test. PWWP2A is highlighted in green, members of the core NuRD (M1HR) complex in red, previously identified H2A.Z-mononucleosome binders[12] in blue, PWWP2A-specific interactors not found in H2A.Z pulldowns[12] in black and background binding proteins in gray. **b** Immunoblots of several NuRD members (MTA1, HDAC2, RBBP7, RBBP4, and CHD4) and H3 upon GST and GST–PWWP2A IP with HK cell-derived mononucleosomes. **c** Upper part: schematic depiction of mammalian MTA1-3 paralogues. Lower part: immunoblots of PWWP2A or MBD3 after IP of endogenously tagged MTA1–FLAG, MTA2–GFP, or MTA3–FLAG from mouse embryonic stem cell (mESC) nuclear extracts. Input lanes represent 10% of the lysate used for the IP. **d** FLAG-PWWP2A IPs with cell lysates from HEK293 cells co-transfected with combinations of plasmids encoding FLAG-PWWP2A, HDAC1 (tagless), HA-RBBP4, and either HA-MTA1 or HA-MTA2. Left panel: western blot of inputs. Right panel: SYPRO Ruby-stained SDS-PAGE of the precipitated proteins. **e** Top: schematic depiction of domain structure of PWWP2A and deletion constructs. Bottom: coomassie-stained SDS–PAGE gel with indicated recombinant PWWP2A deletion constructs on beads (left) and immunoblots of IPs from lysates from HEK293 cells expressing HA-MTA1 (right)

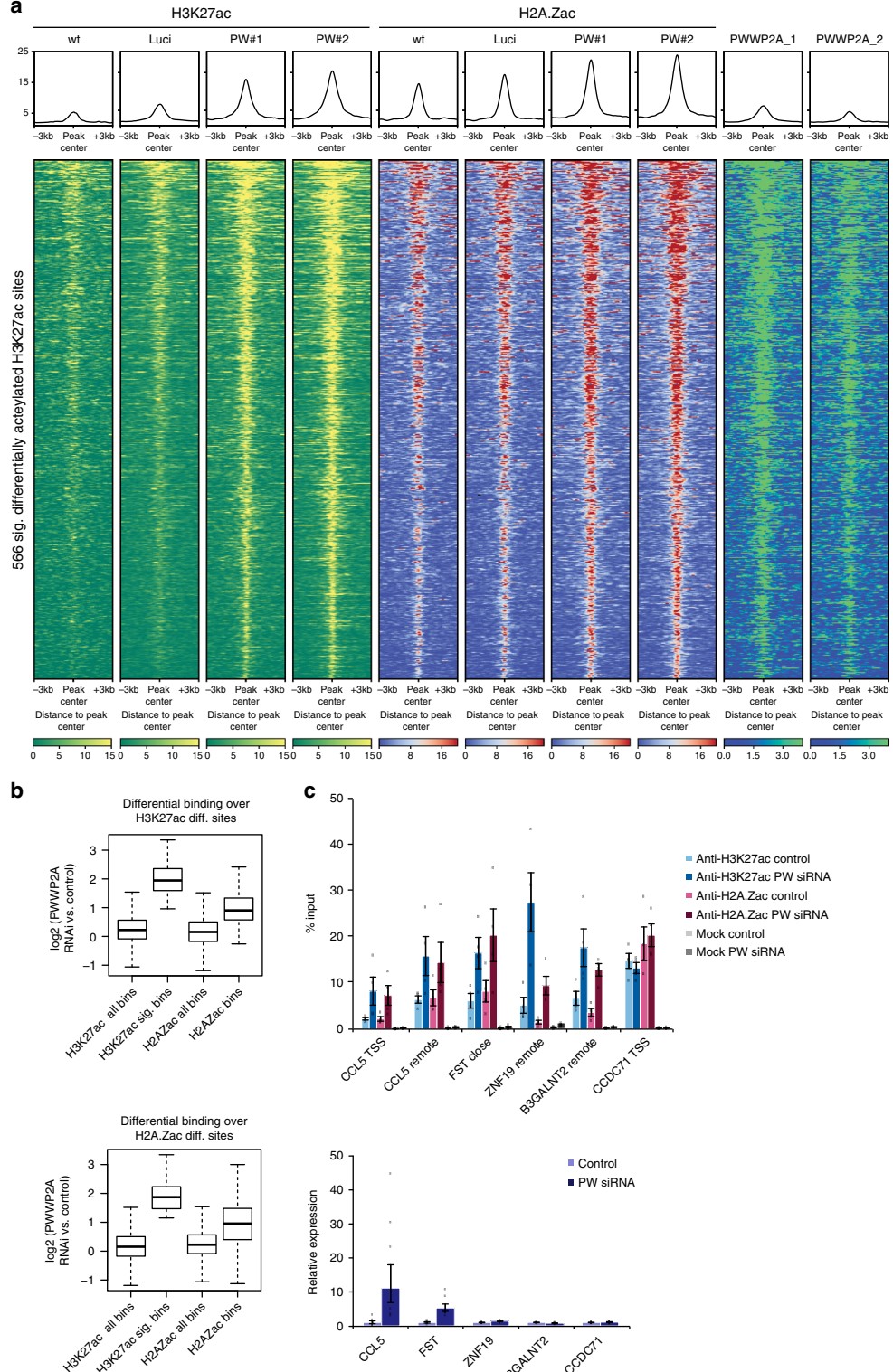

et al., suggest that NuRD might be assembled from two modules: one comprising HDAC, MTA and RBBP subunits in a 2:2:4 ratio and the other comprising a GATAD2, MBD, and CHD subunit[25,35]. The data presented here demonstrate a function for one of these modules in isolation.

We have previously shown that MBD3 can interact directly with an MTA subunit[36], indicating that MBD3 constitutes a bridge between the MHR complex and the CHD remodeling subunit. The data we present here suggest that PWWP2A directly competes with MBD3 for binding to MTA1, providing a mechanistic explanation for the lack of enrichment of MBD, GATAD2, and CHD proteins in the PWWP2A pulldowns. The findings also raise the likely possibility that other proteins can act to decouple the MHR complex from the remainder of NuRD.

The preference of PWWP2A for binding M1HR over M2HR, which we could recently also confirm in another independent

**Fig. 6** Depletion of PWWP2A mediates increase of H3K27 and H2A.Z acetylation. **a** nChIP-seq density heatmap of 566 induced H3K27ac sites upon PWWP2A-depeletion. H3K27ac (yellow) and H2A.Zac (red) mark intensities at these regions upon control (wt, Luci) or PWWP2A (PW#1, PW#2) siRNA-mediated knockdown. Color intensity represents normalized and globally scaled tag counts. Notice strong signal intensity increase of H3K27ac and H2A. Zac upon PWWP2A depletion at PWWP2A-bound regions (green, two replicates are shown). **b** Boxplots showing quantification of binding events at 566 sites differentially acetylated at H3K27 ($p < 0.001$) after siRNA-mediated knockdown of PWWP2A (top). Plots indicate that the majority of the observed acetylation changes were characterized by induction of acetylation ($p$ value: sig. H3K27ac versus all: $p < 2.2e{-}16$). These sites show simultaneous induction of H2A.Zac, although not as strongly pronounced as for H3K27ac ($p$ value $< 2.2e{-}16$) (top). Analysis of differentially regulated H2A.Zac (423 sites, bottom) indicates prevalent induction of acetylation ($p$ value $< 2.2e{-}16$). These sites show simultaneous induction of H3K27ac, although not as strongly pronounced as for H2A.Zac ($p$ value $< 2.2e{-}16$). Boxplots represent the median and first and third quartiles with whiskers indicating the most extreme data point that is no more than 1.5 times the length of the box away from the box. **c** Top: validation of nChIP-seq by nChIP-qPCR at selected loci. Shown is percent input of two replicates of H3K27ac or H2A.Zac nChIPs upon control (wt, Luci) or PWWP2A (PW: PW#1, PW#2) siRNA-mediated knockdown. Error bars depict SEM ($n = 4$). Notice two different classes of genes/loci: Increase of acetylation levels on H3K27ac and H2A. Zac can result in (i) upregulation of expression of a nearby gene (CCL5, FST) or (ii) no change in expression of a nearby gene (ZNF19 and B3GALTN2); CCDC71 represents a locus where acetylation level and gene expression remain unchanged. TSS: transcriptional start site, remote: regulatory region close to a gene. Bottom: Relative expression of selected genes corresponding to differentially acetylated sites (upper panel) after control (wt, Luci) or PWWP2A (PW: PW#1, PW#2) siRNA-mediated knockdown. Shown is the fold change of three replicates normalized to HPRT expression. Error bars depict SEM ($n = 6$). Notice upregulation of gene expression of only CCL5 and FST

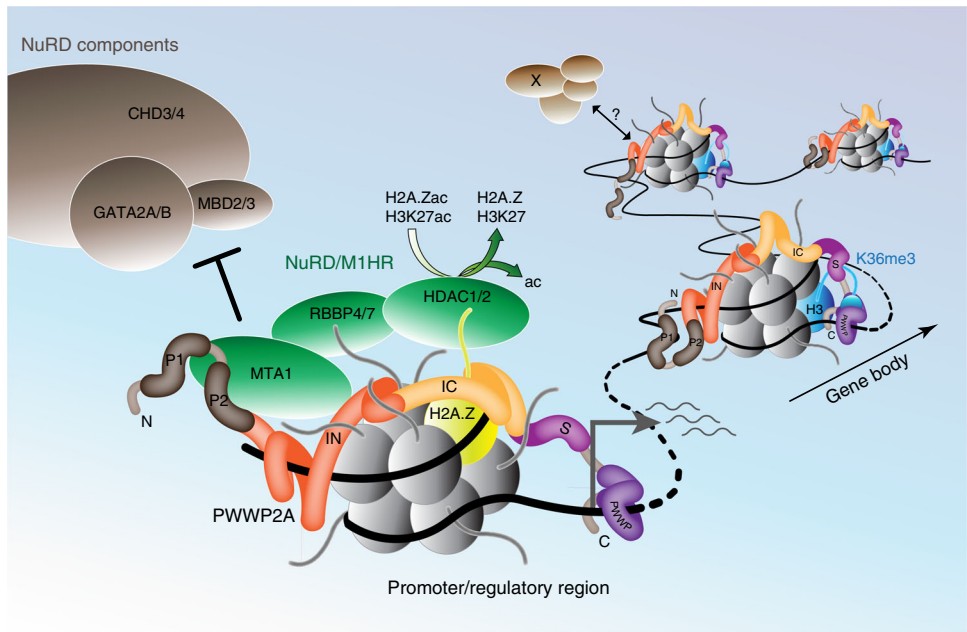

**Fig. 7** Model of PWWP2A recruitment to chromatin and binding to M1HR complex. PWWP2A is able to interact with different chromatin states via distinct domains. It binds weakly to H3K36me3-enriched gene body regions using aromatic cage residues in its S_PWWP domain and recognizes free nucleic acid strands with the PWWP domain. Additionally, PWWP2A binds strongly to H2A.Z-containing promoter and regulatory sites with the IN domain interacting with linker DNA and the IC region mediating H2A.Z–nucleosome specificity. On chromatin, PWWP2A serves as adapter between distinct nucleosome components (H3K36me3 or H2A.Z) and chromatin-modifying complexes. Notably, H2A.Z-bound PWWP2A interacts with a MTA1-specific core NuRD complex (M1HR), preventing formation of full NuRD (+CHD/MBD/GATA subunits) by competing with the MTA-MBD interaction. PWWP2A thereby leads to the recruitment of HDAC1/2 to promoter and regulatory regions, resulting in turn in the deacetylation of nearby H3K27 and H2A.Z. This activity could dampen transcriptional output by reducing acetylation of regulatory regions to prevent hyperacetylation at active genomic sites

study using MTA isoform-specific pulldowns and MS identification[37], was also unexpected and points to another mechanism by which paralogue choice in NuRD has functional consequences. We also note that despite the clear preference for MTA1 in our direct interaction experiments, MTA2 and MTA3 were identified in our purification of cellular PWWP2A. Because the MHR complex contains two copies of an MTA subunit, it is quite possible that either MTA2 or MTA3 can be "brought along for the ride" in an MHR complex containing one copy of MTA1 and one copy of a different MTA—and that PWWP2A interacts only with the MTA1 subunit. This hypothesis remains to be tested.

We hypothesize that depletion of PWWP2A leads to reduced recruitment of M1HR, and that the resulting loss of HDAC

activity is what leads to the observed increase in H3K27 acetylation. Regulation of acetylation levels by HATs and HDACs is important to balance transcriptional output[29]. Because PWWP2A is a strong binder of H2A.Z, we additionally investigated acetylation levels of H2A.Z, which mainly localize at enhancers and promoters of actively transcribed genes[38,39]. Both PTMs, as well as NuRD, are found at active promoters and might play important roles in transcriptional regulation, especially during development[11,40]. Previously, we found ~600 genes to be deregulated upon PWWP2A knockdown[12]. Indeed, we found that a fraction of nearby acetylation sites of upregulated genes shows an increase in acetylation levels. Since it is largely unknown which genes are influenced by which distant regulatory regions it

is difficult to faithfully determine corresponding genes. As changes in chromatin environment go hand-in-hand with transcriptional regulation, the increase in acetylation levels on H3K27 and H2A.Z provides a mechanistic explanation for the observed transcriptional deregulation. It has been suggested that, in mESCs, NuRD acts as a transcriptional modulator that is involved in maintaining bivalency presumably by removing the active H3K27ac mark and consequently promoting H3K27 tri-methylation[11,26,41]. We propose that PWWP2A recruits HDACs in the context of M1HR complex, to transcriptionally active or bivalent H2A.Z-containing regions in order to balance transcriptional output at these sites. Both HDAC 1 and 2 are found at promoters of active genes[29], which fits well with our hypothesis that a core NuRD complex is recruited to H2A.Z-containing promoters of highly transcribed genes via interaction with PWWP2A. Alternatively, PWWP2A–M1HR's primary targets may not be histones but other proteins, such as chromatin modulators and transcription factors (TFs). It is known that the acetylation status of TFs influence their gene regulatory activities[42,43], therefore, loss of PWWP2A and consequent delocalization of a fraction of HDAC1/2 might lead to changes in TF acetylation and explain the observed upregulation as well as downregulation of genes[12].

Studies in differentiation systems, where dynamic chromatin is even more relevant, and development of inducible PWWP2A knockouts and rescue experiments, will help to shed light on the functional connection between PWWP2A and M1HR. Such studies might reveal how the PWWP2A–M1HR complex, and/or the binding of PWWP2A to other complexes regulates mitotic progression and cranial–facial development.

In summary, we show that PWW2A uses at least four different domains to bind different chromatin moieties. We propose that PWWP2A acts to recruit M1HR to H2A.Z-containing chromatin that aids in balancing acetylation states.

## Methods

**Cell culture**. HK cells[12], kind gift from Heinrich Leonhardt (LMU Munich), were grown in Dulbecco's modified Eagle's medium (DMEM, Sigma) supplemented with 10% fetal calf serum (FCS; Sigma) and 1% penicillin/streptomycin (37 °C, 5% $CO_2$). HK cells stably expressing eGFP or eGFP-tagged PWWP2A were previously described[12], and are routinely tested for mycoplasma contamination.

The EXPI293F™ cells are originally from Life Technologies (now ThermoFisher Scientific, Cat no. #A14527). This cell line is derived from the HEK293-F cell line, which was derived from the HEK293 cell line. As for accession numbers, the HEK293-F and HEK293 cell lines have ExPASy accession numbers CVCL_6642 and CVCL_0045, respectively.

Mouse ESCs (mESCs) are derivatives from the E13tg2a parent line and were grown on 0.1% gelatin in 2i/LIF conditions as previously described[44]. Mbd3 floxed cells in which tagged MTA cell lines were made are described in ref. [45]. The epitope tagged mESC lines (Mbd3-Avi-3 × FLAG, Mta1-Avi-3 × FLAG, Mta2–GFP, and Mta3-Avi-3 × FLAG) were made by standard gene targeting methods[37].

**siRNA transfections**. siRNAs were designed using siDESIGN Center (http://dharmacon.gelifesciences.com/design-center/?redirect=true) and synthesized (MWG-Biotech AG). siRNAs were prevalidated by BLAST searches (NCBI) to confirm their targeting specificity to PWWP2A and to reduce the chance of off-target effects. PWWP2A-specific siRNAs were described before[12]. The following double stranded siRNAs were used: Luciferase, 5′-CUUACGCUGAGUACUUC GA-3′; PWWP2A#1, 5′-GGACAGAAGUCAAGUGUGAUUU-3′; PWWP2A#2, 5′-GCUAUUAAACUACGACCCAUUU-3′. Cells were transfected with siRNA using oligofectamine (Invitrogen) according to the manufacturer's instructions. Two days after transfection, cells were harvested for RNA extraction and nChIP experiments.

**Plasmids**. For expression in bacteria, PWWP2A cDNA was cloned into a pGEX-6P1 (Amersham) plasmid and PWWP2A deletions were cloned as previously described[12]. GST-PWWP2A_ΔIC was generated via Gibson Assembly (New England Biolabs) using the following oligos:
F: 5′-CCAGAGTAAGAACTCTGACTCTTCCAGTGC TTC-3′ and

R: 5′-TGGAAGAGTCAGAGTTCTTACTCTGGAGAACTTTTTTAGTAC TTAACTG-3′ single point mutations in pGEX-6P1-S_PWWP were introduced using site-directed mutagenesis with the following primers:
S_PWWP_F666A: 5′-CAAGATATATGGCGCCCCTTGGTGGCCAG-3′
S_PWWP_W669A: 5′-GGCTTCCCTTGGGCGCCAGCCCGTATTC-3′
S_PWWP_W695A: 5′-GAGGCCCGTATTTCAGCGTTTGGGTCTCC-3′
PCR was performed under standard conditions with 20 cycles and the template plasmid DPNI digested. The newly amplified plasmid, containing the desired mutations, was transformed into DH5α (Genentech). Correct sequences of all DNA products established by PCR-induced mutagenesis or cloning were verified by sequencing (MWG).

For expression in human cells, PWWP2A cDNA was cloned into a pIRESneo-EGFP plasmid[12], generated from pIRESneo (EcoRV and BamHI digest) with an EGFP insert from pEGFP-C1 (NheI digest followed by Klenow reaction and BamHI digest).

The pcDNA3.1 expression vector was used for transfection of suspension-adapted HEK Expi293F™ cells. Full-length genes for human PWWP2A (UniProt ID: Q96N64), RBBP7 (UniProt ID: Q16576), mouse MTA2 (UniProt ID: Q9R190), human MTA1 (UniProt ID: Q13330), and human HDAC1 (UniProt ID: Q13547) were cloned with or without N-terminal FLAG- and HA-tags.

**Antibodies**. The following commercially available antibodies were used:
Anti-H3 (1:5.000, #61475), anti-H3K36me3 (1:4.000, #61102), anti-H3K36me2 (1:1.000, #39255), anti-H3K36me1 (1:1.000, #61351), anti-H3K4me3 (1:500, #61379) and anti-H2A (1:1.000, #39209) were from Active Motif. Anti-H3K27me3 (1:1.000, C15410069), anti-H3K79me3 (1:1.000, C15310068), anti-H2A.Zac (2 μg/IP, C15410202-050) and anti-H3K9me3 (1:1.000, C15410056) were from Diagenode. Anti-H4K20me3 (1:1.000, ab9053), anti-H3 (1:30.000, ab1791), anti-MTA1 (1:1.000, ab71153 or 3 μg/IP, ab50209), anti-CHD4 (1:1.000 or 2 μg/IP, ab70469), anti-HDAC2 (1:5.000, ab124974) anti-RBBP4 (1:1.000, ab488), anti-RBBP7 (1:1.000, ab3535), anti-H3K27ac (2 μg/IP, ab4729), anti-H2A.Z (1:3.000, ab4174), and anti-MBD3 (1:5.000, ab157464) were from Abcam. Anti-PWWP2A (1:1.000, NBP2-13833) from Novus (Acris). Anti-HA-HRP (1:40.000, 2999S) and anti-HDAC1 (1:20.000, 5356S) from Cell Signaling Technology. Anti-FLAG M2 (1:80.000 or 3 μg/IP, F1804) and anti-mouse IgG (3 μg/IP, I8765) from Sigma-Aldrich.

The following secondary antibodies were used:
Anti-rabbit IRDye 800CW (1:10.000, 926-32211) and anti-mouse IRDye 680RD (1:10.000 926-68070) from LI-COR Biosciences. Anti-mouse-HRP (1:10.000, M114) from Leinco Technologies. Anti-mouse and anti-rabbit HRP-linked antibodies (1:10.000, NA931 and NA934) from VWR.

**MS identification of histone modifications**. Sample preparation: immunoprecipitated protein-fractions, separated by a 4–20% gradient sodium dodecylsulfate polyacrylamide gel electrophoresis (SDS–PAGE) (SERVA), were stained with Coomassie (Brilliant blue G-250) and protein bands in the molecular weight range of histones (15–23 kDa) were excised as single band/fraction. Gel pieces were destained in 50 % acetonitrile/ 50 mM ammonium bicarbonate. Lysine residues were chemically modified by propionylation (30 min, RT) with 2.5 % propionic anhydride (Sigma) in ammonium bicarbonate (pH 7.5) to prevent tryptic cleavage. This step only added a propionyl group to unmodified and monomethylated lysines, whereas lysines with other side chain modification will not obtain an additional propionyl group. Subsequently, protein digestion (200 ng of trypsin (Promega)) in 50 mM ammonium bicarbonate was performed (overnight (ON)) and the supernatant was desalted by C18-Stagetips (reversed-phase resin) and carbon Top-Tips (Glygen) according to the manufacturer's instructions. Following carbon stage tip, the dried peptides were resuspended in 17 μl of 0.1 % TFA.

LC-MS analysis of histone modifications: 5 μl of each sample were separated on a C18 home-made column (C18RP Reposil-Pur AQ, 120 × 0.075 mm × 2.4 μm, 100 Å, Dr. Maisch, Germany) with a gradient from 5 % B to 30 % B (solvent A 0.1 % FA in water, solvent B 80 % ACN, 0.1 % FA in water) over 32 min at a flow rate of 300 nl/min (Ultimate 3000 RSLC Thermo-Fisher, San Jose, CA) and directly sprayed into a Q-Exactive HF mass spectrometer (Thermo-Fisher Scientific). The mass spectrometer was operated in the PRM mode to identify and quantify specific fragment ions of N-terminal peptides of human histone H3.1 and histone H4 proteins. In this mode, the mass spectrometer automatically switched between one survey scan and 9 MS/MS acquisitions of the m/z values described in the inclusion list containing the precursor ions, modifications and fragmentation conditions (Supplementary Data 1).

Survey full scan MS spectra (from m/z 250–800) were acquired with resolution 30,000 at m/z 400 (AGC target of $3 \times 10^6$). PRM spectra were acquired with resolution 15,000 to a target value of $2 \times 10^5$, maximum IT 60 ms, isolation window 0.7 m/z and fragmented at 27% normalized collision energy. Typical mass spectrometric conditions were: spray voltage, 1.5 kV; no sheath and auxiliary gas flow; heated capillary temperature, 250 °C.

Quantification of histone modifications: Data analysis was performed with Skyline (version 3.6)[46] by using doubly and triply charged peptide masses for extracted ion chromatograms (XICs). Peaks were selected manually and the integrated peak values (Total Area MS1) were exported as.csv-file for further calculations. The percentage of each modification within a given peptide is derived from the ratio of this structural modified peptide to the sum of all isoforms of the

corresponding peptides. Therefore, the Total Area MS1 value was used to calculate the relative abundance of an observed modified peptide as percentage of the overall peptide. Coeluting isobaric modifications were quantified using three unique MS2 fragment ions. Averaged integrals of these ions were used to calculate their respective contribution to the isobaric MS1 peak (e.g.,: H3K36me3 and H3K27me2K36me1).

Percentage values of each modification were normalized to percentage values of Input samples and plotted with Perseus software version 1.5.1.6 with euclidean clustering and subsequent visualization in heatmaps[47]. Default settings were used for analysis in Perseus.

**Label-free quantitative MS**. The sample preparation of HK cell-derived mononucleosomes and following label-free quantitative MS were carried out as previously described[12] with the following differences:

LC-MS/MS analysis: Peptides were analyzed by reversed-phase liquid chromatography on an EASY-nLC 1000 or 1200 system (Thermo Fisher Scientific, Odense, Denmark) coupled to a Q Exactive plus or HF mass spectrometer (Thermo Fisher Scientific). HPLC columns of 50 cm length and an inner diameter of 75 µm were in-house packed with ReproSil-Pur 120 C18-AQ 1.9 µm particles (Dr. Maisch GmbH, Germany). Peptide mixtures were separated using linear gradients of 120 or 140 minutes (total run time + washout) and a two buffer system: buffer A + + (0.1% formic acid) and buffer B + + (0.1% formic acid in 80% acetonitrile). The mass spectrometer was operated in a data-dependent top 10 or top 15 mode. Peptides were fragmented by higher energy collisional dissociation (HCD) with a normalized collision energy of 27.

MS Data analysis: MS raw data were processed using the MaxQuant software version 1.4.3.13[48]. Fragmentation spectra were searched against a human sequence database obtained from Uniprot in May 2013 and a file containing frequently observed contaminants such as human keratins. Cysteine carbamidomethylation was set as a fixed modification; N-terminal acetylation and methionine oxidation were set as variable modifications. Trypsin was chosen as specific enzyme, with 2 maximum missed cleavages allowed. Peptide and protein identifications were filtered at a 1% FDR. Label-free quantification was performed using the MaxLFQ algorithm[49] integrated into MaxQuant. The match between runs option was enabled with a matching time window of 0.5 min and an alignment time window of 20 min. All other parameters were left at standard settings. MaxQuant output tables were analyzed in Perseus[47] version 1.5.8.6 as follows: After deleting proteins only identified with modified peptides, hits to the reverse database, contaminants and proteins with one or less razor and unique peptides, label-free intensities were log2 transformed. Next, proteins were required to have 3 valid values in at least one triplicate, then remaining missing values in the data matrix were imputed with values representing a normal distribution around the detection limit of the mass spectrometer. Now a two-sample t test was performed to identify proteins enriched in the PWWP2A pulldowns compared control. Only those proteins were kept for further analysis. Significant outliers were determined using a permutation-based FDR. The S0 and FDR parameters were set to 0.5 and 0.05, respectively (Supplementary Data 2 and 3).

**Recombinant expression and purification of GST proteins**. 50 ng plasmid DNA were transformed into competent E, coli (BL21). For each construct one clone (colony) was picked, inoculated (0.1 mg/ml ampicillin) and again incubated (37 °C, ON) while shaking. Next, pre-culture was inoculated in LB medium and grown until an OD of 600 nm was reached. The pre-culture was induced by adding 0.3 mM IPTG (Roth) and incubated for 16–18 h (18 °C) while shaking. Following, the culture was pelleted and shock frozen in liquid nitrogen and resuspended in lysis buffer (PBS, 0.4 M NaCl, protease inhibitors). After sonication (Branson Sonifier 250-D), cells were centrifuged and the supernatant was mixed with glutathione sepharose beads (GE healthcare). After immunoprecipitation (2 h, 4 °C), beads were washed 3 × 5 min with wash buffer (PBS, protease inhibitors). For storage (−80 °C), beads were resuspended in storage buffer (TE, 50 mM NaCl, 10% glycerol).

Some experiments required eluted protein. Here, GST–protein was purified via a GST-trap column (GE healthcare) and the Äkta pure system (GE healthcare). After elution with elution buffer (PBS, 30 mM glutathione (Sigma)), proteins were dialyzed (4 °C, ON) using a special membrane (MWCO 6-8.000, Spectra/Pore, (Roth)) in 1 L potassium phosphate buffer (PPB) (150 mM NaCl, 25 mM HEPES pH 7.6, 2 mM MgCl$_2$, 10% glycerol, 0.1 mM EDTA, 0.05% NP-40). Purified, GST-tagged proteins were concentrated by using centrifugal filter units (Millipore).

**Preparation of S1 mononucleosomes and immunoprecipitation**. For each IP, S1 mononucleosomes from $1 \times 10^7$ HK cells were prepared as previously described[12]. GST-tagged proteins were washed 2× in washing buffer, added to S1 mononucleosomes and incubated (4 °C, ON, rotating). Following immunoprecipitation, proteins were washed 2× for 10 min in reduced EX-100 I (0.1% NP-40, 10 mM HEPES pH 7.6, 150 or 300 mM NaCl, 1.5 mM MgCl$_2$, protease inhibitors) and 2 × 10 min in reduced EX-100 II (10 mM HEPES pH 7.6, 150 or 300 mM NaCl, 1.5 mM MgCl$_2$, protease inhibitors) buffers. Proteins were eluted by adding 50 µl Laemmli Sample Buffer (95 °C, 5 min), loaded onto SDS–PAGE gels and immunoblotted (Supplementary Fig. 8). Immunoblot detection of proteins was done using fluorescently labeled secondary antibodies and band intensities quantified using a

LI-COR machine (Bioscience). Analysis was performed by normalization to H3 signals, with the exception of GST-control precipitations.

**Recombinant mononucleosome assembly**. Recombinant human H2A, H2A.Z, H2B, H3, and H4 histones were provided by Hataichanok Scherman (www.histonesource.com). Totally, 147 and 187-bp DNA fragments (pUC18, gift of Mueller-Planitz, LMU Munich) were used for mononucleosomes with either 0 or 20 bp linker DNA, respectively. DNA was prepared by PCR using Cy3- or Cy5-labeled primers (Sigma). Resulting PCR products were polyethyleneglycol- and isopropanol precipitated and resolved in 1× TE (pH 8). As previously described[50,51], the assembly of histone octamers and reconstitution of mononucleosomes was performed by salt-gradient dialysis. Glycerol gradient ultracentrifugation was applied to enhance the purity of recombinant mononucleosomes. Finally, the nucleosomal DNA concentration was determined by UV absorption measurement (260 nm).

Recombinant human TL histone octamers were provided by Hataichanok Scherman (www.histonesource.com). Ten microlitre reactions were set up containing 0.85 µg histone octamers and 1 µg DNA in initial dilution buffer (10 mM HEPES pH 7.9, 1 mM EDTA pH 8, 0.5 mM PMSF) with a final salt concentration of 2 M NaCl and incubated at 37 °C for 15 min. Reactions were transferred to 30 °C and serial diluted with initial dilution buffer (3.3, 6.7, 5.0, 3.6, 4.7, 6.7, 10, 30, and 20 µl). Fifteen minute incubation step (30 °C) was done after each dilution. Finally, reaction was diluted in 100 µl final dilution buffer (10 mM Tris-HCl pH 7.5, 5 mM EDTA pH 8, 20% glycerol, 0.1% NP-40, 5 mM DTT, 0.5 mM PMSF, 100 µg/mL BSA fraction V (Sigma)), again incubated at 30 °C for 15 min and stored at 4 °C.

**Competitive electrophoretic mobility shift assay**. Depending on the application, 5% (recombinant mononucleosomes) or 8% (nucleic acids) native polyacrylamide gels (0.2× TBE, x% polyacrylamide, 0.9% APS, 0.1% TEMED) were used and prerun (1 h, 100 V, ice bath) in running buffer (0.2× TBE).

Recombinant, GST-tagged proteins were diluted to desired concentrations in PPB buffer (150 mM NaCl, 25 mM HEPES pH 7.6, 2 mM MgCl$_2$, 10% glycerol, 0.1 mM EDTA, 0.05% NP-40). Cy5-labeled R-601 DNA (25 bp) (Eurofins Genomics) or Cy3-labeled R-601 RNA (25 bp) (Sigma) and recombinant mononucleosomes (Cy3 or Cy5) were diluted in EMSA binding buffer (TE, 50 mM NaCl, 10% glycerol, 0.1% BSA, 0.5 mM DTT) to a final concentration of 20 nM and mixed 1:1 with different protein concentrations and incubated for 25 min on ice (dark) in a total volume of 15 µl per reaction. Samples were loaded on a native gel and ran (1 h 45 min, 100 V, dark) on ice and analyzed using a Typhoon FLA9500 (GE Healthcare Life Sciences).

**Immunoprecipitation and immunoblotting of MTA1 and MBD3**. Nuclear extract was prepared using a modified version of previously described procedures[45,52]. Briefly, cells were resuspended in hypotonic Buffer A (10 mM HEPES-KOH, pH 7.9, 1.5 mM MgCl$_2$, 10 mM KCl) with 0.5 mM dithiothreitol (DTT) and protease inhibitors and incubated on ice for 15 min. NP-40 was added at final concentration of 0.6% and cells were vortexed to obtain cell nuclei. After centrifugation of cell lysate, nuclear pellet was resuspended in Buffer C (420 mM NaCl, 20 mM HEPES-KOH, pH 7.9, 20% v/v glycerol, 2 mM MgCl$_2$, 0.2 mM EDTA) with 0.1% NP-40, 0.5 mM DTT and protease inhibitors. The suspension was incubated with rotation for 1 h at 4 °C, and then spun at 18,000×g for 20 min at 4 °C. The supernatant was aliquoted, snap frozen in liquid nitrogen and stored at −80 °C.

For immunoprecipitation, indicated antibodies were incubated with Protein G-Sepharose beads (Sigma) for 1 h at room temperature (RT). Nuclear extract (200 µg) was diluted in IP-buffer (50 mM Tris-HCl, pH 7.5, 150 mM NaCl, 1 mM EDTA, 1% Triton X-100, 10% glycerol) with protease inhibitors and incubated with antibody-bead conjugates at 4 °C ON. The beads were then washed three times in low-salt wash buffer (IP-buffer containing 150 mM NaCl), followed by two washes in high-salt wash buffer (IP-buffer containing 300 mM NaCl). Immunoprecipitated proteins were eluted by boiling in 2× Laemmli sample buffer with DTT.

For Western Blot analysis, proteins were separated by SDS–PAGE and blotted onto nitrocellulose membrane. The membrane was blocked for 1 h in TBS-T containing 5% skimmed dry milk and incubated ON at 4 °C with the indicated antibodies diluted in Blocking Buffer. After washes, membranes were incubated with HRP-conjugated secondary antibodies diluted in TBST for 1 h at RT. HRP activity was detected using ECL Western Blotting Detection Reagent (GE Healthcare).

**Binding of recombinant MTA1 and PWWP2A**. Suspension-adapted HEK Expi293F$^{TM}$ cells were grown to a density of $2 \times 10^6$ cells/ml. Combinations of equimolar quantities of plasmids were co-transfected into cells using linear 25-kDa polyethylenimine (PEI) (Polysciences, Warrington, PA, USA). Six microgram of DNA mix was first diluted in 324 µL of PBS and vortexed briefly. Twelve microgram of PEI was then added and the mixture was vortexed again, incubated for 20 min at RT, and then added to 3.0 mL of HEK cell culture. The cells were incubated for 65 h at 37 °C with 5% CO$_2$ and horizontal orbital shaking at 130 rpm. Totally, 1.5-ml aliquots of cells were then harvested, washed twice with PBS, centrifuged (300g, 5 min), snap frozen in liquid nitrogen and stored at –80 °C.

Lysates were prepared by sonicating thawed cell pellets in 0.5 ml of lysis buffer (50 mM Tris-HCl, 500 mM NaCl, 1% (v/v) Triton X-100, 1× cOmplete® EDTA-free protease inhibitor (Roche, Basel, Switzerland), 0.2 mM DTT, pH 7.9), incubating on ice for 30 min to precipitate chromatin and then clarifying the lysate via centrifugation (≥16,000g, 20 min, 4 °C). The cleared supernatant was used for FLAG-affinity pulldowns as described below.

**Immunoprecipitation of proteins produced in HEK293 cells**. In all, 20 µl of anti-FLAG Sepharose 4B beads (Bimake, Houston, TX, USA; pre-equilibrated with 50 mM Tris-HCl, 500 mM NaCl, 1% (v/v) Triton X-100, 1× cOmplete® EDTA-free protease inhibitor (Roche), 0.2 mM DTT, pH 7.9) was added to 0.5 mL of cleared HEK cell lysate. The mixtures were incubated for 3 h at 4 °C with orbital rotation. Postincubation, the beads were washed with 3× 1 ml "wash" buffer A (50 mM Tris-HCl, 500 mM NaCl, 0.5% (v/v) IGEPAL® CA630, 0.2 mM DTT, pH 7.5), and 2× 1 ml "wash" buffer B (50 mM Tris-HCl, 150 mM NaCl, 0.5% (v/v) IGEPAL® CA630, 0.2 mM DTT, pH 7.5). Bound proteins were eluted by 3× 20 µl treatment with "elution" buffer (10 mM HEPES, 150 mM NaCl, 300 µg/ml 3× FLAG peptide (MDYKDHDGDYKDHDIDYKDDDDK)), pH 7.5 for 1 h at 4 °C. Elution fractions were pooled for downstream analyses.

**SDS–PAGE and immunoblot analysis**. Samples of Expi293F™ lysates and anti-FLAG purification protein elutions were subjected to protein gel electrophoresis using Bolt™ 4–12% Bis–Tris Plus gels (Thermo Fisher Scientific) and run in MES buffer at 165 V for 50 min. Gels were washed, fixed and stained with SYPRO Ruby (Thermo Fisher Scientific) according to the manufacturer's instructions. The gels were scanned with a Typhoon FLA-9000 laser scanner (473 nm excitation, ≥510 nm emission filter; GE Healthcare, Chicago, IL, USA). For western blot analysis, the gel-separated proteins were blotted onto PVDF membranes and probed with appropriate antibodies (Supplementary Fig. 9).

**Recombinant production of S_PWWP and NMR spectroscopy**. Escherichia coli Rosetta (DE3) pLysS cells transformed with an S_PWWP encoding plasmid (or point mutants) were cultured at 37 °C with shaking. Log-phase cultures (OD: 0.4–0.8) were induced by addition of IPTG (0.3 mM), and cultured for a further 18 h at 18 °C with shaking. Cell pellets containing overexpressed protein were lysed by snap freezing followed by sonication and purified using GSH affinity chromatography. The N-terminal GST-tag was removed by incubation of GSH elution fractions with HRV-3C protease (produced in-house) at 4 °C. Fractions were concentrated in a centrifugal concentrator to 50–200 µM and one-dimensional ¹H NMR spectra recorded on a Bruker 600 MHz NMR spectrometer at 25 °C.

**Native ChIP-seq (nChIP-seq)**. Preparation of samples for nChIP-seq was performed as previously described[12] with a few adjustments. In brief, S1 mono-nucleosomes isolated from $5 \times 10^6$ HK cells subjected to PWWP2A or control siRNAs were used per nChIP. Fifteen microlitre magnetic Dynabeads Protein G (Thermo Fisher) were precoupled for 2.5 h with either 2 µg anti-H3K27ac or anti-H2A.Zac and incubated with mononucleosomes ON. Beads-only IP served as negative control. Beads were washed twice with WB1 (10 mM Tris pH 7.5, 1 mM EDTA, 0.1% SDS, 0,1% sodiumdeoxycholate, 1% TritonX-100), twice with WB2 (150 mM NaCl, 10 mM Tris pH 7.5, 1 mM EDTA, 0.1% SDS, 0.1% sodiumdeoxycholate, 1% Triton X-100), once with TE + 0.2% Triton-X and once with TE buffers. Washed beads were resuspended in 100 µl TE, 3 µl 10% SDS and 5 µl of 20 mg/ml proteinase K and incubated for 1 h at 65 °C. Suspensions were vortexed briefly, magnetically separated and the supernatant transferred to a fresh tube. Beads were washed once with 100 µl TE containing 0.5 M NaCl, magnetically separated and the supernatant mixed with the first supernatant. Input fractions were processed in parallel. After Phenol/chloroform/isoamylalcohol extraction and ethanol precipitation, the DNA pellet after IP was resuspended in 12 µl and the input DNA pellet in 32 µl 10 mM Tris-HCl (pH 7.5). DNA concentrations were determined with the Qubit dsDNA HS Kit (Invitrogen) and DNA size monitored on a 1000 DNA BioAnalyzer chip (Agilent). Illumina Sequencing libraries were established with the MicroPlex Library Preparation Kit (Diagenode) following the manufacturer's instructions. The number of amplification cycles was scaled according to the amount of input material and amplification was validated by determination of DNA concentrations using the Qubit dsDNA HS Kit (Invitrogen). Quality of purified libraries were assessed using a 1000 DNA BioAnalyzer chip (Agilent) . Sequencing was performed on the Illumina 1500 platform using the TruSeq Rapid SR Cluster Kit-HS (Illumina) and single read 50 nucleotide sequencing on a HiSeq Rapid SR Flow Cell (Illumina).

Image analysis and base calling were performed using the Illumina pipeline v 1.8 (Illumina Inc.). Raw and processed data have been deposited in the NCBI gene expression omnibus (GEO) under accession number GSE110222.

**Public data sets analyzed**. H3K4me3, PWWP2A, H2A.Z.1 and H2A.Z.2 data from HeLa Kyoto cells were used as previously deposited at GEO (GSE78009). ChIP-seq data for additional histone modifications were downloaded from the ENCODE portal at UCSC (http://hgdownload.soe.ucsc.edu/goldenPath/hg19/encodeDCC/wgEncodeBroadHistone/).

H4K4me1: wgEncodeBroadHistoneHelas3H3k04me1StdRawDataRep1.fastq.gz;
H3K4me3: wgEncodeBroadHistoneHelas3H3k4me3StdRawDataRep1.fastq.gz;
H3K27ac: wgEncodeBroadHistoneHelas3H3k27acStdAlnRep1.fastq.gz;
H3K27me3: wgEncodeBroadHistoneHelas3H3k27me3StdRawDataRep1.fastq.gz;
H3K36me3: wgEncodeBroadHistoneHelas3H3k36me3StdRawDataRep1.fastq.gz.

Raw sequencing reads were mapped to the human genome (hg19) using bowtie (1.1.2) with parameters –k and –m set as 1[53]. PCR duplicates were removed with samtools rmdup function. Coverage vectors were generated after read extension to 150 bp corresponding to the expected fragment size after MNase digestion using Deeptools bamCoverage function normalizing to 1× sequencing depth (–normalizeTo1× option)[54]. Peak calling was done using MACS2 version 2.1.1[55] with default settings. Cumulative coverage plots and heatmaps were generated with deeptools computeMatrix and plotHeatmap functions. In order to generate the Venn diagram in Supplementary Fig. 6a, we included normalized (using DESeq2 method) read counts introducing an additional criterion beyond overlap of detected peak intervals in order to increase stringency. In case of absence of an overlap for a given peak interval, we asked to have at least a twofold enrichment in normalized read counts of the binding factor over the presumptive nonbinding factor (as described in ref. [56]). Thereby we substantially reduced the large number of incidences, were comparison of peak calling results in detection of binding events seemingly unique for one factor, where there actually is substantial binding detected in case of the other factor, too, but the peak calling process just failed to include it in the final peak list. As gene models, we used UCSC hg19 gene models downloaded from Illumina's iGenome repository. For collection of binding data around meta-genes, 6 kb upstream and downstream regions were collected as 50-bp bins and gene bodies were represented by 120 bins of variable size. K36me3-rich gene bodies were identified by collecting. Differential binding analysis was done using csaw[57], and validated by a custom analysis pipeline. For csaw we binned the human genome into 100 bp bins with a step size of 300 bp. Normalization was performed by extraction of scaling factors based on 10 kb bins basically normalizing for total read count. Differentially regulated bins were selected as those bins with a p value < 0.001. For representation of significantly changed regions in heatmaps we merged neighboring significantly changed bins by a sliding window approach (width: 2 kb). In an alternative approach, we calculated the peaks overlapping between control treatments as well as those overlapping between RNAi treatments. Peaks were combined and overlapping peaks were merged into single features spanning the original features' intervals. Read counts overlapping these intervals were determined for the four experimental conditions (untreated, Luci siRNA, PWWP2A siRNA 1 and 2) and DESeq2 was used for normalization as well as for the identification of differentially bound peak intervals[58].

In general, we found good agreement between both methods and in the heatmap in Fig. 6a we used those genomic regions as reference point, that were detected by both algorithms.

**Chromatin states**. In order to assign chromatin states, we used publicly available H3K4me1/3, H3K27ac, H3K27me3, and H3K36me3 data from HeLa cells generated by the ENCODE consortium[59]. We discretized the human genome into 1 kb bins and trained a 10-state HMM using the ChromHMM application using default parameters[60]. In order to calculate the ratio between observed and expected overlap we calculated the overlap between PWW2PA and H2A.Z.1 peaks and the set of intervals associated with the respective chromatin state. This was divided by the sum of all PWWP2A (or H2A.Z.1) peak interval widths in order to calculate the observed overlap. The expected overlap was calculated as the ratio between the sum of the interval widths of the respective chromatin state and the total genome size.

**Plotting and statistics**. All downstream analysis was done in R/BioConductor[61]. Genome browser snap shots were generated using the Gviz package[62]. Manipulation of sequencing reads was done using Rsamtools[63] and genomic intervals were represented as GenomicRanges objects[64]. The analysis of the association between peak intervals and known genomic annotation feature were done using the ChIPseeker package[65] with default setting using the UCSC hg19 gene definitions (BioConductor package TxDb.Hsapiens.UCSC.hg19.knownGene). As statistical tests, we performed Wilcoxon rank sum tests. The code underlying our analysis is available upon request.

**nChIP-qPCR**. QPCR was performed on a LightCycler® 480 Instrument II (Roche) using Fast SYBR Green Master Mix (Applied Biolabs). PCR efficiency and primer pair specificity was examined using a standard curve of serially diluted DNA and melting curve, respectively. Data were analyzed according to the percent input method. The following primer sets were used:
CCL5 TSS:
F: 5′ ATTTCTCTGCTGACATCCTTAGT 3′
R: 5′ TCCTAACTGCCACTCCTTGT 3′
CCL5 remote:
F: 5′ GGATCCCAAGAGAAGCCTGA 3′
R: 5′ CAGGGGCAAAGAAGGAGAGA 3′
FST close:
F: 5′ TGCCATCCTTAGACCTCAGA 3′
R: 5′ AGCACTGCCAGGACTACATT 3′

ZNF19 remote:
F: 5′ CACACAAAACATTTCTCCATCAGG 3′
R: 5′ AGGTTTTCTTGCCTTGGAACA 3′
B3GALNT2 remote:
F: 5′ TGTCTCCCTGAAACTCATCTCT 3′
R: 5′ GCACTAATCCTGCCTTCCTG 3′
CCDC71 TSS:
F: 5′ GTGGTGCATTGACATCTGGG 3′
R: 5′ CTGGGACTGAAGTGGGCATT 3′

**RNA isolation, cDNA preparation, and RT-qPCR.** Samples for RNA isolation were harvested concurrent with samples for nChIP. Total RNA from HeLa Kyoto cells untreated, subjected to control siRNA (Luci) or PWWP2A siRNAs was isolated using the RNeasy Mini Kit (Qiagen) according the manufacturer's instructions. One microgram of total RNA was reverse transcribed utilizing the NEB ProtoScript M-MuLV First Strand cDNA Synthesis Kit (NEB) according the manufacturer's instructions. QPCR was performed on a LightCycler® 480 Instrument II (Roche) using Fast SYBR Green Master Mix (Applied Biolabs). PCR efficiency and primer pair specificity was examined using a standard curve of serially diluted cDNA and melting curve, respectively. After normalization to the transcript level of HPRT1 (hypoxanthine phosphoribosyltransferase 1), data were analyzed based on the $2^{-\Delta\Delta CT}$ method. For incorporating the standard error of the mean of the $\Delta\Delta$CT values into the fold-difference, we calculated the range of the error bars as $2^{(-\Delta\Delta Ct + SEM)}$ and $2^{(-\Delta\Delta Ct - SEM)}$. The following primer sets were used:
CCL5:
F: 5′ CTCGCTGTCATCCTCATTGC 3′
R: 5′ TACTCCTTGATGTGGGCACG 3′
FST:
F: 5′ TGCCTGCCACCTGAGAAAG 3′
R: 5′ TCTCCCAACCTTGAAATCCCA 3′
ZNF19:
F: 5′ CCCAGCAGAGAGGACCAAAA 3′
R: 5′ GCTGACCATGTGACATCATCC 3′
B3GALNT2:
F: 5′ TGGCTGCCATAGGACCTAA 3′
R: 5′ TCCACAGTTCCGTCAGTTCC 3′
CCDC71:
F: 5′ AAAGCTGCTGAAGTTCCGTG 3′
R: 5′ TGGAGCCGTATTACAGGTGA 3′

## Data availability

The nChIP-seq data analyzed in this publication have been deposited in NCBI's Gene Expression Omnibus[66] and are accessible through GEO Series accession number GSE110222. [GSE110222]. The mass spectrometry proteomics data sets have been deposited to the ProteomeXchange Consortium via the PRIDE[67] partner repository with the dataset identifiers PXD010202 [PXD010202] for GFP–PWWP2A IPs, and PXD010424 [PXD010424] for histone PTMs.

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

## Acknowledgments

We thank Ann-Sophie Giel, Merve Kaya, Felix Mueller-Planitz, Christelle Ngueda, Sabrina Pfennig, and Michaela Smolle for technical help and advice. This work was supported by the Deutsche Forschungsgemeinschaft through the Collaborative Research Center SFB1064 (project A10 to S.B.H. and project A16 and Z03 to A.I.) and TRR81 (project A15 to S.B.H.), by CIPSM to S.B.H. and M.M., and the Weigand'sche Stiftung to S.B.H. T.S. and M.B. were supported by TRR81. J.P.M. was funded by a Senior Research Fellowship (APP1058916) and a grant (APP1126357) from the Australian National Health and Medical Research Council. T.B. and B.H. were funded through a Wellcome Trust Senior Fellowship, the EU FP7 Integrated Project 4DCellFate and core funding to the Cambridge Stem Cell Institute from the Wellcome Trust and Medical Research Council. SLi and RMMS are fellows of the Integrated Research Training Group (IRTG) of the SFB1064. S.P. was a fellow of the Munich International Max Planck Research School for Molecular and Cellular Life Sciences (IMPRS-LS).

## Author contributions

S.Li., R.M.M.S. and S.B.H. conceived of this study. R.M.M.S. performed all in vitro binding studies with advice of C.R. S.Li. and S.P. performed human GFP–PWWP2A mononucleosome-IP experiments for LFQ-MS identification conducted by E.C.K. and supported by M.M. S.Li. generated RNAi experiments and nChIP-seq data that were sequenced by A.N. with support of T.S. S.Li. performed nChIP-qPCR and RT-qPCR experiments with help of M.B. M.C.V.A. performed histone PTM mass spectrometry with help of A.I. All bioinformatic analyses were performed by M.B. S.P. generated in silico model of PWWP domain. T.B. performed mouse MTA-pulldown experiments with support of BH. NMR spectroscopy and MTA1 binding studies were carried out by M.S., I.L., J.K.K.L., M.T. and J.P.M. S.B.H., M.B. and S.Li. wrote the manuscript with help of RMMS and support from all other coauthors.

## Additional information

**Competing interests:** The authors declare no competing interests.

