## [Peer Review File · Nature Communications]

Reviewer #1 (Remarks to the Author):

In this manuscript the authors provide detailed biochemical and functional analysis of a the histone reader protein PWWP2A. the authors provide extensive analysis of the interaction of purified portions of PWWP2A with nucleosomes of varying properties concluding that specific regions of PWWP2A interact with DNA and with the histone components of nucleosomes - specifically histone H3 trimethylated at lysine 36. The authors validate this biochemical data via reanalysis of previously published data. The authors go on to define nuclear binding partners of PWWP2A, concluding that a specific module of the NuRD complex comprised of HDAC1/2, MTA1 and RBBP4/7 interacts directly and specifically with PWWP2A - in a manner that excludes this NuRD module from productive interaction with the MBD2/3, GATAD2A/B, CHD3/4 module. Functional analysis in cells leads the authors to conclude that PWWP2A directs HDAC1/2 to specific loci leading to local deacetylation of H3K27.

While I find the overarching concepts here to be meritorious and interesting, I find many details of the manuscript to be puzzling or repetitious with previously published work.

Major Issues:

1. I personally find the first two figures, supplementary Figure 1 and parts of S2 to be confusing and difficult to follow. I do not understand why the upper panels in figure 1C differ so much in banding pattern from panel to panel. by my reading of the text and legend, these should be identical. Have I missed something or is there that much variability in the assay? Please explain.

It is likewise difficult to understand (and the authors do not provide any explanation) why some panels have a single shifted nucleosomal band while others have multiple bands. What do the authors think is happening?

2. The data in Figure 2c seem to me to indicate that the S-PWWP construct has absolutely no modification specificity whatsoever. I do not agree with the authors conclusions here.

3. figures 3 and 4 seem to me to be extremely similar to figures in the Punzeler et al EMBOJ paper (ref 12 in the current work). I do not comprehend the intellectual advance provided here. Perhaps the authors could clearly convey what is new.

4. the IP mass spec are clearly described and provide exciting new avenues to follow - I rather liked this data. I am more than a bit confused as to the meaning and interpretations of the co-IP experiments presented in 5B, C, D. It seems likely to me that MTA proteins will be unstable in the absence of their HDAC partners given the elegant structural data from Schwabe and colleagues.

I do not see where the data presented in 5B support the authors conclusions that PWWP2A co-IPs this NuRD module. the data are extremely weak and this is a critical point in the manuscript.

Likewise the 'identification' of interaction of PWWP2A with both the BAH and C-terminal regions of MTA1 (Supp Fig 5) seems puzzling given that the best interaction observed is with a construct lacking the BAH domain. I do not agree with the conclusions drawn here.

5. figure 6 is likely very important data and merits further exploration. The authors depict approx. 500 regions defined as peaks of PWWP2A. this seems like a very small number. Are the authors confident in their nChIP with GFP?

It would be very helpful to understand the overlap of PWWP2A with H3K27Ac peaks genome wide. Likewise, where are these 500 odd loci relative to genes? Are closely linked genes regulated by depletion of PWWP2A? I urge the authors to more fully explore this dataset.

Reviewer #2 (Remarks to the Author):

Understanding the mechanism of how H2A.Z regulates gene expression and other aspects of genome function is poorly understood but requires elucidating its interacting partners. Previously, this laboratory identified PWWP2A as a specific interacting protein. This important manuscript has characterised the in vitro targets of this multi-domain protein and most interestingly, identified H3K36me3 as a new interaction partner via the PWWP domain. In cells, this interaction with H3K36me3 appears to occur in the absence of H2A.Z (but see below). More significantly, this study has revealed that PWWP2A binds to a specific NuRD-HDAC-containing sub complex leading to increased levels of H3K27 and H2A.Z acetylation. Clarifying and addressing the following issues I believe will strengthen the manuscript.

1. In gel shift assays shown in Fig. 1, multiple gel-shifted complexes are observed when linkerless nucleosomes are used, which are not observed for linker-containing nucleosomes. This was not commented on. This may in part be due to, unfortunately, the efficiency of nucleosome reconstitution. It is obvious that the efficiency of reconstitution varies significantly between experiments with some experiments having no free DNA while in Fig. 1e, reconstitution was low with more free DNA than that reconstituted into a nucleosome. Examining Fig. 1e, GST-IC seems to be binding this free DNA and the H2A.Z nucleosome core equally as well.

2. Along these lines, it stated that IN binds better to H3 TL mononucleosomes compared to wt. mononucleosomes, this is not evident in Fig. 1C so I would suggest showing densitometer scans.

3. Concerning the observation that PWWP2A associates weakly with H3K36me3 in the gene body, it is necessary to show this distribution mapped to the intron-exon boundary. H3K36me3 – containing nucleosomes are more abundant on exons than introns and therefore this relationship should be clearer to establish. Even better, if it is possible to rank both H3K36me3 and PWWP2A with the level of expression. It has been shown that H2A.Z is found on the intron-exon boundary of

inactive genes. Therefore, producing such profiles for H2A.Z is also important. Indeed, a speculative proposal is that H2A.Z recruits PWWP2A to exons on inactive genes. Subsequently, it is maintained at exons by H3K36me3 following transcriptional activation. In addition, it is necessary to produce a similar cluster analysis for H2A.Z as shown in Fig. 3A.

4. The description of the results in this section on page 8 is somewhat confusing. The different states are discussed followed by a brief description of H2A.Z clusters. It is unclear how the different states link to the H2A.Z clusters. Is it possible to add the H2A.Z mark to the different chromatin states in Fig. 4A as an important comparison? Moreover, H2A.Z and PWWP2A are found in all clusters so the specificity with the different histone modifications is not apparent. Along these lines, how does H3K27ac recruit PWWP2A? It seems that for all chromatin states, it is dependent upon H2A.Z and independent of the mark e.g. H3K27ac. Therefore, this analysis seems to be more about which marks H2A.Z is associated with rather than any specific interaction with other histone modifications.

5. Concerning the interaction between PWWP2A and the MHR sub complex, it is uncertain why PWWP2A was able to pull down only MTA1 from the cell lysate and not HDAC1 and RBBP4 when they are all part of a stable complex.

6. It would be valuable to correlate the loss of PWWP2A and the increase in acetylation with changes in gene expression to provide more support to the argument that PWWP2A is involved in regulating genes expression.

7. It would also be informative to perform a clustering analysis to determine the percentage overlap between those genomic regions that increase in H3K27ac with those regions that increase in H2A.Z. If there are regions that increase in H3K27ac in the absence in H2A.Z, how do the authors propose this H2A.Z independent increase in acetylation (point 4 above)?

8. How can these new observation explain the involvement of PWWP2A in mitotic progression and cranial-facial development.

Reviewer #3 (Remarks to the Author):

In the manuscript, the authors studied PWWP2A binding different chromatin moieties and interacting with proteins. Previously they found that PWWP2A tightly binds to H2A.Z-containing nucleosomes and is involved in mitotic progression. In this work, first, they tested different domains of PWWP2A mediating the binding to H2A.Z, free linker DNA and H3K36me3 nucleosomes using in vitro assays. In vivo, they found that PWWP2A strongly recognized H2A.Z-containing regulatory regions and weakly bound to H3K36me3-containing gene bodies. Combining IP with MS-based

proteomics, they observed that PWWP2A bound to an MTA1-specific sub-complex of the NuRD complex, consisting of MTA1, HDAC1 and RBBP4/7, termed as M1HR, but not CHD, GATA2 and MBD. Depletion of PWWP2A resulted in the acetylation increase on H3K27 and H2A.Z, which was assumed to be related to impaired chromatin recruitment of M1HR. This work is interesting, and novel results about PWWP2A interactions with chromatin and other proteins are presented, which help us have a better understanding of gene regulation.

On Page 8: "PWWP2A binding was strongly enriched at active promoters, which are characterized by high levels of H3K4me3 and H3K27ac (state 3)." In Fig. 2C, when they tested the binding between different domains (also some combination of the domains) of PWWP2A and H3K4me3, no any interactions were observed with H3K4me3. It might be better to give some explanations.

It is interesting that S_PWWP strongly binds to H3K36me3, but the interaction is very weak between the protein PWWP2A or I_S_PWWP and H3K36me3. Whether might it be possible to test the interactions between the protein with the removal of the I domain and H3K36me3, which will make the statement of "an element in the I domain might auto-inhibit the S_PWWP domain" more convincing.

Regarding the pulldown experiments, non-specific binding is often an issue. Normally triplicate (at least duplicate) experiments are required to increase the confidence for interactor identifications. In addition, in Fig 5A, there are several grey dots with high $-\log P$ and T-test difference. What are those proteins?

Minor point:

In the second paragraph on Page 14: "We hypothesize that depletion of PWWP2A leads to reduced recruitment of M1HR, and that the resulting loss of HDAC activity is what leads to the observed decrease in H3K27 acetylation." Here it should be increase in H3K27 acetylation, instead of "decrease".

RE: Resubmission of manuscript NCOMMS-18-04252 (Link & Spitzer et al.)

Dear Referees,

we have recently received your reviews for our manuscript NCOMMS-18-04252. First, we wish to thank you for your time and thoughtful feedback. We were happy to see that you found the manuscript of potential interest for publication in *Nature Communications*.

We have, as requested, deposited all proteomics data sets to a community repository (ProteomeXchange Consortium via the PRIDE partner repository) with the following reviewer account details:

GFP-PWWP2A IPs:

Project accession: PXD010202

Username: reviewer11653@ebi.ac.uk

Password: EQ9mPRoj

Histone PTMS:

Project accession: PXD010424

Username: reviewer78944@ebi.ac.uk

Password: kxYkRLKG

In addition, we have highlighted all changes made to the manuscript in red.

Below find our point-by-point responses to the comments:

Responses to Referee #1:

Major issues

1. I personally find the first two figures, supplementary Figure 1 and parts of S2 to be confusing and difficult to follow. I do not understand why the upper panels in figure 1C differ so much in banding pattern from panel to panel. by my reading of the text and legend, these should be identical. Have I missed something or is there that much variability in the assay? Please explain.

It is likewise difficult to understand (and the authors do not provide any explanation) why some panels have a single shifted nucleosomal band while others have multiple bands. What do the authors think is happening?

We understand the referee's concerns that the multiple bands in Figure 1c can be confusing. For more clarity and to emphasize our findings, we exchanged Figures and added quantifications of biological replicates. In detail, we moved Figure 1c to the Supplement (linker-less mononucleosomes, now Supplementary Figure 1d), and replaced it by Supplementary Figure 1d (20-bp linker-containing mononucleosomes, now Figure 1c). Additionally, we have added a quantification of signal intensities of the cEMSA's shown in Figure 1c and Supplementary Figure 1d and 1f. In the experiment shown in initial Figure 1c free DNA is present, which leads to non-distinguishable and unassignable band patterns. Since IN is known to interact with DNA, also free 147-bp DNA is shifted in addition to linker-less mononucleosomes, causing, dependent on its concentration, differences in band patterns within distinct panels. In the new Figure

1c almost no free 187-bp DNA is present and shifted bands correspond to mononucleosomes complexed with GST-IN. Here, mononucleosome band-shifts can be distinguished. At the highest protein concentration (80 nM), two shifted bands are present, indicating that each mononucleosome can presumably be bound by two GST-IN proteins as two DNA linkers are available. Moreover, it has been reported that the H3 tail interacts with linker DNA, thus it is conceivable that tail-less H3 mononucleosomes with linker DNA are bound better by IN, because there is no competition between the H3 tail and GST-IN for linker DNA binding. Interestingly, when no linker DNA is present IN binds slightly better to H4 TL nucleosomes (Supplementary Fig. 1d). As the H4 tail has been shown to bind to H2A's acidic patch, it is possible that in addition to the more affine DNA binding ability of IN, it also recognizes, to a lesser extent, regions in H2A surrounding the acidic patch. We have changed the text accordingly.

Supplementary Figure 2 shows the binding properties of GST-PWWP to different nucleic acids and mononucleosomes. The PWWP domain can bind mononucleosomes without any preference for canonical H2A or H2A.Z. However, PWWP binds better to mononucleosomes in the presence of linker DNA and the two upshifted bands also suggest that two PWWP can bind to one nucleosome simultaneously.

We generally agree with the referee that EMSA experiments can be complicated to interpret and that the different binding modes of PWWP2A's distinct domains to different chromatin moieties we are investigating are rather complex. Therefore, we have quantified the results of many replicates (new quantifications to Figure 1c and Supplementary Figure 1d and f) and added for clarification reasons and to summarize the different binding properties of PWWP2A a new summary table (Supplementary Figure 4a).

2. The data in Figure 2c seem to me to indicate that the S-PWWP construct has absolutely no modification specificity whatsoever. I do not agree with the authors conclusions here.

We do not agree with the referee in this point. When looking closely at the immunoblots and the quantification of the immunoprecipitation signals in Figure 2c (right and bottom), it is clearly observable that S_PWWP strongly binds to nucleosomes enriched in H3K36me3. This experiment has been independently repeated three times, quantified and revealed the same result in each case: strong enrichment of H3K36me3 nucleosomes in S_PWWP IPs (strong H3K36me3 signal in S_PWWP IP compared to input and IPs with other PWWP2A constructs; Figure 2c bottom, red bar). As this result was also verified by MS analyses, we are confident that S_PWWP recognizes H3K36me3-enriched nucleosomes.

3. figures 3 and 4 seem to me to be extremely similar to figures in the Punzeler et al EMBOJ paper (ref 12 in the current work). I do not comprehend the intellectual advance provided here. Perhaps the authors could clearly convey what is new.

In our previous paper (Pünzeler et al., EMBO J., 2017), we focused mainly on promoter regions, as these showed a strong enrichment in PWWP2A binding in native ChIP-sequencing (nChIP-seq). Now, based on the newly acquired *in vitro* binding assay data in this manuscript, we are extending our bioinformatics analyses to gene bodies (H3K36me3-enriched) and enhancer regions (characterized by H3K4me1 and H3K27ac). Possible localizations of PWWP2A to these genomic features have not been analyzed in detail previously and are therefore new and enhance our understanding of how PWWP2A is able to recognize different chromatin moieties.

4. the IP mass spec are clearly described and provide exciting new avenues to follow - I rather liked this data. I am more than a bit confused as to the meaning and interpretations of the co-IP experiments presented in 5B, C, D. It seems likely to me that MTA proteins will be unstable in the absence of their HDAC partners given the elegant structural data from Schwabe and colleagues.

The referee is correct that MTA1 (or MTA2) forms a dimer of dimers with HDAC1 (or HDAC2). We have found in our work that MTA1 can be expressed in the absence of an HDAC partner and still bind to other partners. Nevertheless, to clarify this point and to address the referee's concern, we have removed Figures 5b, c and d and have replaced them with a single, new panel (new Figure 5d) that we hope is much clearer and emphasize our finding that i) PWWP2A specifically binds MTA1 and not other MTA isoforms and ii) MTA1 is needed for recruitment of the M1HR complex to PWWP2A. This new figure shows FLAG-PWWP2A pulldowns carried out with cell lysates from HEK293 cells co-expressing FLAG-PWWP2A, HDAC1 (untagged), HA-RBBP4 and either HA-MTA1 or HA-MTA2. Only in the presence of MTA1, but not MTA2, PWWP2A pulled down all components of the MHR complex, suggesting that MTA1 is the direct binding partner of PWWP2A and mediates the recruitment of HDAC and RBBP proteins to form an MTA1-specific "M1HR" module. Details on structural aspects of the PWWP2A-M1HR complex go beyond the scope of this manuscript and will be determined in future studies.

I do not see where the data presented in 5B support the authors conclusions that PWWP2A co-IPs this NuRD module. the data are extremely weak and this is a critical point in the manuscript.

We agree with the referee that these were not the most conclusive gels we've ever run. We have repeated the experiment shown in the right-hand panel of the old Figure 5d, which is now shown in the new Figure 5d. These data, as described above, show more clearly that MTA1 is the direct target of PWWP2A and needed for recruitment of the complete M1HR complex.

Likewise the 'identification' of interaction of PWWP2A with both the BAH and C-terminal regions of MTA1 (Supp Fig 5) seems puzzling given that the best interaction observed is with a construct lacking the BAH domain. I do not agree with the conclusions drawn here.

We have repeated these experiments a number of times and find, overall, that the binding of the BAH domain construct is not that strong (certainly less strong than the binding of the construct *lacking* the BAH domain, for example) and is also not as consistent as we would like it to be. Consequently, we have opted to remove this piece of data from the manuscript and will work separately to reach a conclusion that we are happy with. The main conclusions of the paper remain unchanged.

5. figure 6 is likely very important data and merits further exploration. The authors depict approx. 500 regions defined as peaks of PWWP2A. this seems like a very small number. Are the authors confident in their nChIP with GFP?

It would be very helpful to understand the overlap of PWWP2A with H3K27Ac peaks genome wide. Likewise, where are these 500 odd loci relative to genes? Are closely linked genes regulated by depletion of PWWP2A? I urge the authors to more fully explore this dataset.

Concerning the first comment of the referee, we would like to clarify that these approximately 500 regions are not just PWWP2A bound (there are more than 80,000 PWWP2A bound genomic regions, see also new Supplementary Figure 6), but these particular sites are increased in H3K27 and H2A.Z acetylation upon PWWP2A depletion. Why not all PWWP2A regions show these modification changes, we do not know. Maybe not all ~80,000 PWWP2A sites are bound by M1HR that can affect histone acetylation, but are possibly bound by other PWWP2A-

interacting proteins and complexes (see Figure 5a).

Concerning the quality of our nChIP approaches with GFP, we are confident with our data sets. First, the experimental nChIP procedure has been carefully setup and established and was used for many different approaches such as nChIP-MS/MS, nChIP-immunoblotting and nChIP-seq for different proteins (Vardabasso et al., Mol Cell, 2015 and Pünzeler et al., EMBO J., 2017). The GFP-tag should not affect incorporation and localization, as we could show that ChIP-seq with anti-GFP of GFP-H2A.Z cells generates the same genomic localization pattern as ChIP-seq with an antibody against the endogenous H2A.Z (Vardabasso et al., Mol Cell, 2015). Further, nChIP-MS/MS has identified all known binding partners of H2A.Z in addition to new ones, which we have independently confirmed, such as PWWP2A, but also others that have not been published yet. Furthermore, we have verified several of the ~500 differentially acetylated sites by nChIP-qPCR and are therefore confident with our experimental approach.

As the referee suggested, we have now analyzed the overlap of PWWP2A (84.343 peaks), H3K27ac (85.023 peaks) and H2A.Zac (87.482 peaks) and found a genome-wide overlap of 68.457 peaks (72.171 peaks shared by H3K27ac and PWWP2A) (new Supplementary Fig. 6a and b). Interestingly, although about a third of PWWP2A, H3K27ac and H2A.Zac are found at promoters, the ~500 differentially acetylated loci are mostly found at distal intergenic regions (most likely enhancers/silencers) and much less at promoters (new Supplementary Fig. 6c), suggesting that modification changes do not occur randomly but are happening at defined regions and are most likely tightly regulated and functionally important.

We agree with the referee that the investigation of a possible correlation between acetylation changes and gene expression differences upon PWWP2A depletion is of high interest. Therefore we have performed several new experiments that are now shown in new Figure 6b and c, as well as Supplementary Figure 6 and 7. First, we have verified changes in histone acetylation levels upon PWWP2A-depletion on several of the ~500 H3K27ac-/H2A.Zac/PWWP2A-marked regulatory regions by nChIP-qPCR (new Figure 6c and Supplementary Figure 7). Our new results clearly show that the observed increase in H3K27ac and H2A.Zac at regulatory regions upon PWWP2A knockdown is robust and reproducible. Second, we have reanalyzed existing RNA-seq data sets of control and PWWP2A-depleted HeLa cells (Pünzeler et al., EMBO J., 2017) and tried to correlate observed gene expression changes with regions of PWWP2A-depletion induced H3K27 and H2A.Z acetylation increases. However, this analysis was extremely difficult, as most of these differentially acetylated sites are located within distal regulatory regions and not in gene-associated promoters (see new Supplementary Figure 6c). As we are unable to reliably assign genes to their responsible enhancer regions and vice versa, it was not possible to establish a genome-wide and statistically sound correlation analysis. We therefore chose a different approach and decided to look at expression changes of some few genes that are in the direct vicinity of differentially acetylated regions (maybe promoters) by RT-qPCR. These analyses can, of course, not be expanded to reach any generalized conclusions, but show some interesting results. Noteworthy, we found that some genes nearby a differentially acetylated site are indeed upregulated (CCL5, FST) while other remained unchanged in their expression level (B3GALTN2, ZNF19) (new Figure 6c). It is not clear or obvious, what other additional features contribute to gene activation, but we speculate that some genes are more prone to changes in gene acetylation than others. Maybe the genomic location of the regulatory region has an influence, or, alternatively, acetylation changes regulate other more distant genes. More future work on chromatin architecture changes upon PWWP2A depletion will be necessary, but we think that these experiments are beyond the scope of our manuscript. In the focus of this manuscript is the detailed investigation of PWWP2A's multivalent chroma-

tin binding abilities and the first-time demonstration of the interaction of PWWP2A with a core NuRD complex (M1HR) and its effects on genomic histone acetylation.

Responses to Referee #2:

1. In gel shift assays shown in Fig. 1, multiple gel-shifted complexes are observed when linkerless nucleosomes are used, which are not observed for linker-containing nucleosomes. This was not commented on. This may in part be due to, unfortunately, the efficiency of nucleosome reconstitution. It is obvious that the efficiency of reconstitution varies significantly between experiments with some experiments having no free DNA while in Fig. 1e, reconstitution was low with more free DNA than that reconstituted into a nucleosome. Examining Fig. 1e, GST-IC seems to be binding this free DNA and the H2A.Z nucleosome core equally as well.

We agree with the referee that multiple gel-shifted complexes are observed in cEMSAs (please see also our response to point 1 from referee #1). These complexes were observed when mononucleosomes without linker were used and are most likely due to the quality of the nucleosome reconstitution procedure. We have now replaced Figure 1c (linker-less mononucleosomes, now Supplementary Figure 1d) with Supplementary Figure 1d (20-bp linker-containing mononucleosomes, now Figure 1c) to make it as easy as possible for the reader and have added quantifications of the unbound nucleosome bands (**). Further, we have now commented on this fact in the text on page 5, where we speculate that the presence of two shifted bands might be a result of the binding of two GST-IN proteins to both linker DNAs of one mononucleosome. In addition, we have added a new summary table to clarify our findings (new Supplementary Figure 4a).

Regarding the binding of GST-IC in Figure 1e, the new quantifications clearly show that GST-IC binds equally well to linker-less and linker-containing mononucleosomes suggesting no free DNA is necessary to achieve binding.

2. Along these lines, it stated that IN binds better to H3 TL mononucleosomes compared to wt. mononucleosomes, this is not evident in Fig. 1C so I would suggest showing densitometer scans.

To clarify the results, we have now added quantifications to Figure 1c (now Supplementary Figure 1d) and Supplementary Figure 1d (now Figure 1c). It has been reported that the H3 tail interacts with linker DNA, thus it is conclusive that tail-less H3 mononucleosomes with linker DNA are bound better by IN, because there is no competition between the H3 tail and GST-IN for linker DNA binding. Interestingly, when no linker DNA is present, IN binds slightly better to H4 TL nucleosomes (Supplementary Fig. 1d). As the H4 tail has been shown to bind to H2A's acidic patch, it is possible that in addition to the more affine DNA binding ability of IN, it also recognizes, to a lesser extent, regions in H2A surrounding the acidic patch. We have changed the text accordingly. Please see also our comment to point 1 of referee #1.

3. Concerning the observation that PWWP2A associates weakly with H3K36me3 in the gene body, it is necessary to show this distribution mapped to the intron-exon boundary. H3K36me3 –containing nucleosomes are more abundant on exons than introns and therefore this relationship should be clearer to establish. Even better, if it is possible to rank both H3K36me3 and PWWP2A with the level of expression. It has been shown that H2A.Z is found on the intron-exon boundary of inactive genes. Therefore, producing such profiles for

H2A.Z is also important. Indeed, a speculative proposal is that H2A.Z recruits PWWP2A to exons on inactive genes. Subsequently, it is maintained at exons by H3K36me3 following transcriptional activation. In addition, it is necessary to produce a similar cluster analysis for H2A.Z as shown in Fig. 3A.

We agree with the referee that the analysis of PWWP2A at H3K36me3 marked exon-intron boundaries is interesting. Hence, we have reanalyzed our data sets according to previous published analyses (Soboleva TA et al., PLoS Genet. 2017). In order to rank exons according to gene expression levels, we took advantage of our previously published RNA-seq data sets (Puenzeler et al., EMBO J., 2017). In a first attempt, we investigated the complete set of all exons and assigned exons into 4 groups depending on their relative expression levels ordered from non-expressed (top), via lowly to medium and highly expressed genes (bottom) (Rebuttal Figure R1a). In contrast to H2A.Z that is rather depleted, we noticed an enrichment of PWWP2A in exons of highly expressed genes, very similar (although not as pronounced) to that seen for H3K36me3. In addition, we analyzed the exons of genes belonging to cluster 2 (Figure 3A), which previously was selected due to its high H3K36me3 binding signals across gene bodies. As a control group, we defined exons from the same number of the remaining genes (Rebuttal Figure R1b). We were still able to detect a slight enrichment of PWWP2A on exons of cluster 2 versus those of the control group. While performing these analyses, we noticed that we have to take PWWP2A's strong signals at H2A.Z-containing promoter regions into account and adapted our strategy. In order to circumvent this situation, we deleted for all genes the first exon from the analysis (Rebuttal Figure R1c). This approach allowed us to remove a large proportion of the stronger PWWP2A binding signals seen in both clusters. Still, we were able to detect a slight PWWP2A enrichment across the group of cluster 2 exons. Nevertheless the observed enrichment appears not to be as specifically localized to intron-exon boundaries as observed for H3K36me3.

Rebuttal Figure R1: PWWP2A is not as strongly enriched at exon-intron boundaries as observed for H3K36me3. Exon-centered cluster analysis of PWWP2A (two replicates: PWWP2A_1 and _2), H2A.Z.1, H2A.Z.2 and H3K36me3 regions with **A)** using exons of all genes grouped by gene expression levels (top: not expressed, bottom highest expression), **B)** exons of all genes belonging to cluster 2 from Figure 3A (top) and the exons of a control group of the same number of genes coming from the remaining genes (bottom). **C)** Similar to B) but ignoring the first exons.

Concerning the cluster analysis for H2A.Z as shown in Figure 3a, we apologize for causing any confusion. Figure 3a shows clustering of H3K36me3 in order to identify those regions that correspond to gene bodies associated with high levels of H3K36me3. Especially cluster 2 genes show the expected pattern (H3K36me3 distributed over the gene body region) and were therefore used for further analyses that also included H2A.Z.1 and H2A.Z.2 (see Figure 3b).

4. The description of the results in this section on page 8 is some what confusing. The different states are discussed followed by a brief description of H2A.Z clusters. It is unclear how the different states link to the H2A.Z clusters. Is it possible to add the H2A.Z mark to the different chromatin states in Fig. 4A as an important comparison? Moreover, H2A.Z and PWWP2A are found in all clusters so the specificity with the different histone modifications is not apparent. Along these lines, how does H3K27ac recruit PWWP2A? It seems that for all chromatin states, it is dependent upon H2A.Z and independent of the mark e.g. H3K27ac. Therefore, this analysis seems to be more about which marks H2A.Z is associated with rather than any specific interaction with other histone modifications.

Concerning the description of the results on page 8 we have rephrased this text passage to make it easier to understand (page 9). In addition we have now added, as suggested by the referee, H2A.Z to our chromatin states analysis and changed Figure 4b accordingly. As expected, both PWWP2A and H2A.Z show a very similar state distribution, suggesting that both proteins are functionally correlated, not only at promoters but also at enhancer regions. This observation is nicely confirmed by the heatmap analysis in Fig. 4c, indicating that PWWP2A is present at promoters (clusters 1 and 2) as well as active and poised enhancers (clusters 3 and 4) together with H2A.Z.

Furthermore, we hypothesize that PWWP2A is not directly recruited by H3K27ac to chromatin but rather binds to H2A.Z-containing nucleosomes that also contain acetylation marks, such as H3K27ac and H2A.Zac. In order to investigate whether histone acetylation has any influence on PWWP2A binding to nucleosomes, we have increased histone acetylation levels by treatment of cells stably expressing GFP, GFP-H2A, GFP-H2A.Z.1 or GFP-H2A.Z.2 with the histone deacetylase inhibitor trichostatin A (TSA) and performed GFP-mononucleosome pull-downs (see Rebuttal Figure R2). Interestingly, increased acetylation levels on H3K27 did not enhance binding of PWWP2A to H2A.Z, while, as expected, H2A.Z pulldown of bromodomain containing 2 (BRD2), a known acetyl and H2A.Z co-binding protein (Draker R. et al., PLoS Genet., 2012), was strongly increased. Hence, PWWP2A binding to nucleosomes is unaffected by the histone acetylation status and rather depends on the presence of histone variant H2A.Z, free linker DNA and, to a lower extend, H3K36me3.

Rebuttal Figure R2: PWWP2A binding to H2A.Z nucleosomes is unaffected by histone acetylation levels. Immunoblots of BRD2, PWWP2A, GFP or H3K27ac upon IP with GFP-Trap using mononucleosomes derived from HeLa K cells stably expressing GFP, GFP-H2A, GFP-H2A.Z.1 or GFP-H2A.Z.2 untreated or treated with HDAC inhibitor trichostatin A (TSA). Notice that binding of PWWP2A to H2A.Z nucleosomes is not enhanced upon TSA treatment.

5. Concerning the interaction between PWWP2A and the MHR sub complex, it is uncertain why PWWP2A was able to pull down only MTA1 from the cell lysate and not HDAC1 and RBBP4 when they are all part of a stable complex.

In fact, if we co-express PWWP2A and MTA1 in HEK293 cells, we do observe some endogenous HDAC1 and RBBP4 being pulled down (as noted above in response to point 4 of referee #1). This does not affect the conclusions of the experiments described in the new Figure 5d because the differences in band intensities are clear for the overexpressed subunits.

6. It would be valuable to correlate the loss of PWWP2A and the increase in acetylation with changes in gene expression to provide more support to the argument that PWWP2A is involved in regulating genes expression.

We agree with the referee that the investigation of a possible correlation between acetylation changes and gene expression differences upon PWWP2A depletion is of high interest. Therefore we have performed several new experiments that are now shown in new Figure 6b, c and Supplementary Figures 6 and 7. Please see a detailed explanation in our response to point 5 of referee #1. While genome-wide correlation analyses of differentially acetylated regions with gene expression changes were not possible due to our inability to functionally connect enhancers with their regulated genes, we found some transcriptional changes of genes marked by increased acetylation in the direct vicinity by RT-qPCR (new Figure 6c). However, not all genes we investigated showed such an increase and it is not clear or obvious, what other additional features contribute to gene activation. We speculate that some genes are more prone to changes in gene acetylation than others. Maybe the genomic location of the regulatory region has an influence, or, alternatively, acetylation changes regulate other more distant genes. More future work on chromatin architecture changes upon PWWP2A depletion will be necessary, but we think that these experiments are beyond the scope of our manuscript. In the focus of this manuscript is the detailed investigation of PWWP2A's multivalent chromatin binding abilities and the first-time demonstration of the interaction of PWWP2A with a core NuRD complex (M1HR) and its effects on genomic histone acetylation.

7. It would also be informative to perform a clustering analysis to determine the percentage overlap between those genomic regions that increase in H3K27ac with those regions that increase in H2A.Zac. If there are regions that increase in H3K27ac in the absence in H2A.Z, how do the authors propose this H2A.Z independent increase in acetylation (point 4 above)?

We thank the referee for this instructive suggestion and have now included such an analysis (new Figure 6b). Indeed, there is a clear correlation between increases in H3K27ac as well as

H2A.Zac. Regions identified as differentially acetylated at H3K27ac are likewise induced for H2A.Zac and vice versa.

8. How can these new observation explain the involvement of PWWP2A in mitotic progression and cranial-facial development.

This is indeed a very interesting question that we are currently not able to answer. We can think of several possible explanations that need to be experimentally addressed in future studies and go beyond the scope of this manuscript. One possibility is that indeed gene expression changes contribute to the observed phenotypes upon PWWP2A depletion and that these transcriptional alterations are, at least in part, due to M1HR (HDAC) influenced changes in histone acetylation. But, one has to keep in mind that our MS/MS analyses revealed several other PWWP2A interacting proteins, of which many are also found in H2A.Z pulldowns, while some others seem to be PWWP2A-specific. As most of the other binding factors are also able to influence or act on chromatin structure, and some have been shown to influence normal brain development (reviewed in Garay et al., Epigenomics, 2016), it is likely that they all contribute, to a different extent, to mitotic progression and cranial-facial development. We have added a speculative sentence on this topic in the discussion section.

Responses to Referee #3:

1. On Page 8: “PWWP2A binding was strongly enriched at active promoters, which are characterized by high levels of H3K4me3 and H3K27ac (state 3).” In Fig. 2C, when they tested the binding between different domains (also some combination of the domains) of PWWP2A and H3K4me3, no any interactions were observed with H3K4me3. It might be better to give some explanations.

We agree with the referee that the signal of the H3K4me3 immunoblot in Figure 2c was very faint and have now exchanged the blot to clarify this issue. In our new blot, PWWP2A-bound nucleosomes are clearly enriched with H3K4me3, most likely due to the interaction with promoter located H2A.Z nucleosomes.

2. It is interesting that S_PWWP strongly binds to H3K36me3, but the interaction is very weak between the protein PWWP2A or I_S_PWWP and H3K36me3. Whether might it be possible to test the interactions between the protein with the removal of the I domain and H3K36me3, which will make the statement of “an element in the I domain might auto-inhibit the S_PWWP domain” more convincing.

This is a very interesting point. Using a PWWP2A construct with the IC domain deleted (GST-PWWP2A_ΔIC) in recombinant GST-IPs with cell-derived mononucleosomes we did observe an increased binding to H3K36me3 while binding to H2A.Z was reduced (new Figure 2d). Therefore, we are convinced that PWWP2A’s loss of H2A.Z binding contributes to an enhanced interaction with H3K36me3. We have added a description of this new data set in the text (pages 7 and 12).

3. Regarding the pulldown experiments, non-specific binding is often an issue. Normally triplicate (at least duplicate) experiments are required to increase the confidence for interactor identifications. In addition, in Fig 5A, there are several grey dots with high $-\log P$ and T -test difference. What are those proteins?

GFP-PWWP2A pull-downs with quantitative label-free MS/MS were performed twice (see also Supplementary Fig. 5b) and showed high reproducibility (we are using three technical replicates in each independent experiment). These types of analyses were tested vigorously in previous studies (Vardabasso et al., Mol Cell, 2015 and Pünzeler et al., EMBO J., 2017) to determine reproducibility and number of technical and biological replicates needed for high confidence. We determined that two biological replicates with three technical replicates were sufficient. The grey dots are proteins that were only found in one of the two independent experiments and might constitute low-affinity binders or background interactions. We added a corresponding explanatory sentence to the legend of Figure 5a.

Minor point:

In the second paragraph on Page 14: “We hypothesize that depletion of PWWP2A leads to reduced recruitment of MIHR, and that the resulting loss of HDAC activity is what leads to the observed decrease in H3K27 acetylation.” Here it should be increase in H3K27 acetylation, instead of “decrease”.

We thank the referee for his/her attentiveness and have corrected our mistake accordingly in the text.

In sum, we have reworked many parts of our manuscript and included many additional data sets to essentially address all of the concerns and hope that you look favorably on a final decision regarding our manuscript.

Thank you for your time and attention.

Sincerely,

Sandra B. Hake

Reviewer #1 (Remarks to the Author):

I thank the authors for thoughtful revision.

Reviewer #2 (Remarks to the Author):

The authors have addressed all my concerns and I recommend the manuscript be accepted.

Reviewer #3 (Remarks to the Author):

With the changes based on the previous comments, the manuscript has been improved. In my view, the responses and changes are satisfactory.

RE: Resubmission of manuscript NCOMMS-18-04252 (Link & Spitzer et al.)

Dear Referees,

We have recently received your latest reviews for our manuscript NCOMMS-18-04252 and want to thank you for supporting the publication of our story in *Nature Communications* without requesting any further experiments.

Sincerely,

Sandra B. Hake